# 📌 Personalized Instance-based Navigation Toward User-Specific Objects in Realistic Environments

**Luca Barsellotti*** **Roberto Bigazzi***
**Marcella Cornia** **Lorenzo Baraldi** **Rita Cucchiara**
University of Modena and Reggio Emilia, Italy
`{firstname.lastname}@unimore.it`

Project page: aimagelab.github.io/pin

## Abstract

In the last years, the research interest in visual navigation towards objects in indoor environments has grown significantly. This growth can be attributed to the recent availability of large navigation datasets in photo-realistic simulated environments, like Gibson and Matterport3D. However, the navigation tasks supported by these datasets are often restricted to the objects present in the environment at acquisition time. Also, they fail to account for the realistic scenario in which the target object is a user-specific instance that can be easily confused with similar objects and may be found in multiple locations within the environment. To address these limitations, we propose a new task denominated *Personalized Instance-based Navigation* (PIN), in which an embodied agent is tasked with locating and reaching a specific personal object by distinguishing it among multiple instances of the same category. The task is accompanied by 📌 *PInNED*, a dedicated new dataset composed of photo-realistic scenes augmented with additional 3D objects. In each episode, the target object is presented to the agent using two modalities: a set of visual reference images on a neutral background and manually annotated textual descriptions. Through comprehensive evaluations and analyses, we showcase the challenges of the PIN task as well as the performance and shortcomings of currently available methods designed for object-driven navigation, considering modular and end-to-end agents.

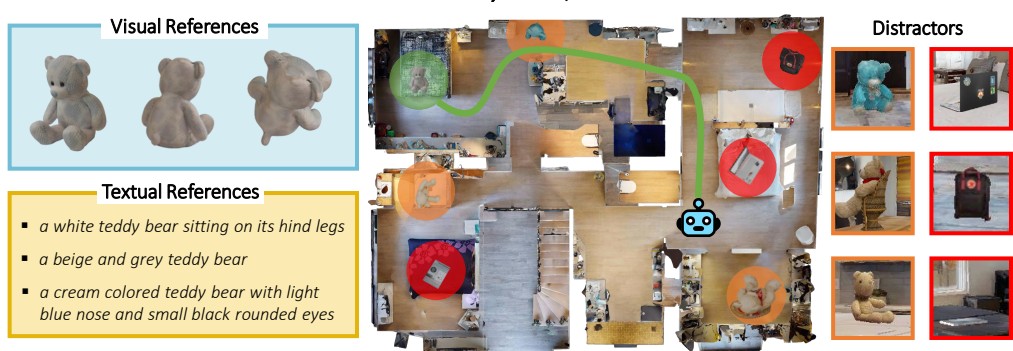

Figure 1: We introduce the PIN task, where the agent is asked to navigate toward a personalized object instance using multimodal references and distinguish it from distractors (*i.e.*, other objects of the same category as the target or of other categories). The target object, same category distractors, and other distractors are circled, respectively, in green, orange, and red. The total number of available objects in the dataset is 338, corresponding to different instances of 18 object categories.

---

*Equal contribution.

38th Conference on Neural Information Processing Systems (NeurIPS 2024) Track on Datasets and Benchmarks.

# 1 Introduction

Imagine a scenario where your child wants his favorite teddy bear, and he lost it somewhere in your house. In the foreseeable future, a "smart" domestic robot could be asked to find it. In that case, the robot will start roaming through the environment searching for the teddy bear. However, prior knowledge of the object category and visual cues related to the surroundings are not enough to solve the task, as the teddy bear has no predetermined location in the scene, could be potentially situated in several different places, and can be confused with other stuffed toys. While the recent advances in Embodied AI have significantly fostered the development of autonomous agents that can locate predefined target object categories, a benchmark that evaluates how agents tackle the challenges of reaching personal object instances in a photo-realistic environment is absent.

**Motivation.** The majority of current object-driven navigation tasks in Embodied AI define their goals as a general semantic category represented through text [2, 6, 70] (*e.g.*, "chair", "sofa") or as a specific target instance defined by an image or description including the surrounding context in which the object can be found [9, 19, 32, 36, 82]. Moreover, these datasets rely on objects which were present at the time of acquisition of the environment [8, 13, 20, 32, 36, 45, 61, 71, 74]. On the contrary, procedurally generated environments can freely contain additional objects and annotations [21, 23, 34, 41]. However, the appearance discrepancy between these environments and the real world or photo-realistic environments could affect the performance of the agents when deployed on robotic platforms [31]. Previous work has proposed loading additional 3D objects inside photo-realistic environments [46] to improve agent navigation performance, to allow object interaction in static environments [64], or to enable navigation towards multiple goals [70]. However, no previous work has targeted loading objects that can be moved frequently and can appear in multiple contexts since loaded 3D models are kept in their initial spawn position.

**Overview of the dataset.** To overcome these issues, we propose the novel task of *Personalized Instance-based Navigation* (PIN), where the agent needs to locate and reach a specific personalized target instance in the environment provided as reference images and textual descriptions, without information about the surrounding context. An overview of PIN is shown in Fig. 1. In parallel with the definition of the task, we release 📌 PInNED (Personalized Instance-based Navigation Embodied Dataset), a dedicated dataset of episodes for this setting that leverages the main advantages of both photo-realistic and procedurally generated embodied environments. In each episode, along with a unique target instance, distractors objects are placed in the scene to confound the navigation of the agent. Specifically, we built the dataset on top of the semantic annotations [74] and scenes of Habitat-Matterport3D Dataset (HM3D) [56] with the injection of additional photo-realistic 3D objects accurately selected from Objaverse-XL [22]. The objects are positioned in each environment through a procedural spawning method on predefined suitable surfaces. PInNED comprises 865.5k training episodes and 1.2k validation episodes built on top of 338 additional objects.

Finally, we adapt and test currently available navigation agents on the proposed dataset, showcasing the shortcomings of relevant approaches. In particular, we compare the performance of the two main categories of navigation agents for object-driven navigation, modular and end-to-end approaches, where we demonstrate that the versatility of modular methods leads to superior performance compared to the end-to-end counterparts; still, the task is far from being resolved. These experiments assess the difficulties posed by PIN task, highlighting the need for further research on the topic. More details and release information on the codebase for the task, accompanying dataset, and evaluation benchmark are included in the Appendix.

**Contributions.** To sum up, our key contributions are threefold:
- 📌 We introduce the task of Personalized Instance-based Navigation (PIN). In this task, an agent must find and navigate towards a specific object instance without using the surrounding context. To increase the difficulty and compel the agent to learn to identify the correct instance, object distractors belonging to the same or different categories of the target are also added.
- 📌 We build and release Personalized Instance-based Navigation Embodied Dataset (PInNED), a task-specific dataset for embodied navigation based on photo-realistic personalized objects from Objaverse-XL dataset injected in the environments of HM3D dataset. Overall, it comprises 338 object instances belonging to 18 different categories positioned within 145 training and 35 validation environments, for a total of approximately 866.7k navigation episodes.
- 📌 We evaluate currently available object-driven methods on the newly proposed dataset demonstrating their limitations in tackling the proposed PIN task.

## 2 Related Work

**Object-based Embodied Datasets.** In recent years, research aimed at the development of intelligent autonomous agents has acquired increasing interest with the release of simulation platforms like Habitat [53, 61, 66], AI2-THOR [34], RoboTHOR [21], and ProcTHOR [23], as well as datasets of scenes for robotic navigation like Gibson [64, 71], Matterport3D [13], and Habitat-Matterport3D (HM3D) [56]. The evaluation of the capabilities of such agents can be performed on multiple embodied tasks [3, 58, 65] mimicking different real-world requirements. PointGoal Navigation (PointNav) [2] requires the agent to reach specific relative coordinates to its starting position. In object-oriented navigation, the agent is tasked to find any instance of an object category (ObjectNav) [2, 6], multiple objects in sequence (MultiON) [70], or a specific instance of a category (ION) [41]. Other embodied navigation tasks are ImageGoal navigation (ImageNav) [19, 82] that requires the agent to reach the position where the goal image has been taken, and a more object-oriented formulation of ImageNav called Instance-Specific Image Goal Navigation (InstanceImageNav) [36] that requires to reach a precise object instance given a photo of it. Recently, the GOAT-Bench benchmark has been introduced, which requires finding sequences of target objects using multimodal references [32]. However, GOAT-Bench targets are constrained to the objects captured in the environment at acquisition time. To the best of our knowledge, PInNED is the only dataset focused on navigation toward personalized targets that uses multimodal references, injects additional objects into photorealistic environments, and requires the agent to distinguish the correct instance from distractors without relying on context.

**Object-based Navigation Agents.** Object-based methods for navigation agents can be divided into two categories depending on their design: modular approaches and end-to-end approaches. Modular approaches are composed of multiple components, usually a mapping module, an exploration procedure, and an object detection method. Some approaches adapted the architecture proposed by ANS [16] for object goal navigation by building semantic maps to locate the target [15, 39, 55, 81]. Following, Stubborn [44] proposed a strong baseline using a heuristic exploration method. Among end-to-end methods, Mousavian *et al.* [50] and Yang *et al.* [76] worked on improving visual representations, Mayo *et al.* [47] used spatial attention maps, and Ye *et al.* [77] used auxiliary tasks. Other related work leveraged object relation graphs [27, 28, 52]. THDA [46], instead, used 3D scans of objects from YCB dataset [11] to augment the training dataset. Recently, PIRLNav [57] used a two-stage learning strategy, Chen *et al.* [18] used a method based on recursive implicit maps, and OVRL [72, 73] exploited self-supervised visual pretraining to boost agent capabilities. Additionally, zero-shot object goal navigation has been recently explored by ZER [1], ZSON [45], and ORION [20].

**Personalized Instance Recognition.** In recent years, foundation models have revolutionized the Computer Vision field. CLIP [54] learned a multimodal embedding space by performing large-scale contrastive training, demonstrating impressive capabilities in zero-shot classification. DINO [12, 51] is trained with a self-supervised paradigm achieving strong semantic correspondence properties among features [4, 5, 79]. Segment Anything (SAM) [33] has been trained to predict precise class-agnostic masks given a prompt. The feature spaces learned by these models are semantically rich and can be exploited in tasks that involve the recognition of general object categories. However, adapting a model for recognizing personalized objects in images remains an open challenge. For example, SuperGlue [60] leveraged an attention-based graph neural network on the local descriptors extracted with the SuperPoint model [25] to perform image matching and has been used in Mod-IIN [35] and GOAT [14] to tackle the InstanceImageNav task. IEVE [40], instead, proposes an Exploration-Verification-Exploitation framework that combines a segmentation model and a keypoint matcher to recognize distant objects and confirm them when the agent is closer; while PerSAM [80], performed personalized segmentation allowing SAM to localize a user-provided target. In the same setting, SegIc [48] introduced a mask decoder with in-context instructions on top of the dense correspondences from DINOv2 [51], while Matcher [43] leveraged DINOv2 to extract prompts for SAM in a training-free paradigm.

## 3 Personalized Instance-based Navigation

In this section, we outline the Personalized Instance-based Navigation task, highlighting its key characteristics and comparing it to existing embodied tasks. Following, we detail the composition and generation process of the PInNED dataset.

Table 1: Comparison of the different object-driven datasets for embodied navigation, considering the photo-realism of scenes and targets, the availability of additional objects with variable spawn locations, the modalities of the provided references, and whether the dataset is instance-oriented.

| Dataset | Photo-Realistic Scenes | Photo-Realistic Targets | Additional Objects | Visual Reference | Descriptive Reference | Variable Placement | Instance Goal |
|---|---|---|---|---|---|---|---|
| MP3D [13] | ✓ | ✓ | ✗ | ✗ | ✗ | ✗ | ✗ |
| AI2-THOR [34] | ✗ | ✗ | ✓ | ✗ | ✗ | ✓ | ✗ |
| Gibson [71] | ✓ | ✓ | ✗ | ✗ | ✗ | ✗ | ✗ |
| Robo-THOR [21] | ✗ | ✗ | ✓ | ✗ | ✗ | ✓ | ✗ |
| MultiON* [70] | ✓ | ✗ | ✓ | ✗ | ✓ | ✓ | ✗ |
| HM3D [56] | ✓ | ✓ | ✗ | ✗ | ✗ | ✗ | ✗ |
| ProcTHOR | ✗ | ✗ | ✓ | ✗ | ✗ | ✓ | ✗ |
| ION [41] | ✗ | ✗ | ✓ | ✗ | ✓ | ✓ | ✓ |
| THDA [46] | ✓ | ✓ | ✓ | ✗ | ✗ | ✓ | ✗ |
| ZSON [45] | ✓ | ✓ | ✗ | ✗ | ✓ | ✗ | ✗ |
| InstanceImageNav [35] | ✓ | ✓ | ✗ | ✓ | ✗ | ✗ | ✓ |
| ZIPON [20] | ✓ | ✓ | ✗ | ✗ | ✓ | ✗ | ✓ |
| GOAT-Bench [32] | ✓ | ✓ | ✗ | ✓ | ✓ | ✗ | ✓ |
| **PInNED (Ours)** | ✓ | ✓ | ✓ | ✓ | ✓ | ✓ | ✓ |

## 3.1 Task Definition

The PIN task requires the agent to navigate toward a predetermined specific object instance (*e.g.*, "*a yellow backpack with red straps*") in an unexplored environment. Each target object needs to be found in the environment, distinguishing it from multiple distractors of the same category and other objects of different categories. In this setting, the target object can be provided to the agent in two different modalities: (i) as a set of RGB images depicting the target object rendered in an isolated context on a neutral background, and (ii) as a set of textual descriptions of the object instance appearance.

At the beginning of each episode of PIN, the agent is initialized at a random pose $x_0$ in an unseen environment. A single target instance $o^i$ is selected as the goal $g$, such that $g \in C^a \subset O$, where $C^a$ is a set of instances belonging to the same object category and $O$ is the set of all available objects. The goal $g$ is placed in the environment at a position $z$. Additionally, $n$ distinct instances $o^j$ ($o^j \in C^a \wedge i \neq j$) are positioned in the environment, along with $m$ distinct instances $o^k$ ($o^k \in (O \setminus C^a)$). At the end of the episode, the navigation is considered successful if the agent selects the '*stop*' action before the maximum allowed number of timesteps $T$, with an Euclidean distance between the position of the agent at the current timestep $x_t$ and the target position $z$ lower than 1 meter. The action space of the agent for the task is defined by six possible actions, where at each timestep $t$, the action $a_t \in \{ $ '*stop*', '*move ahead*', '*turn left*', '*turn right*', '*tilt up*', '*tilt down*' $\}$.

## 3.2 Comparison with Other Tasks

The proposed task locates itself among PointNav [2], ObjectNav [2, 6], ImageNav [19], and the recently defined task of InstanceImageNav [36]. PIN exhibits similarities to ObjectNav, InstanceImageNav, and the recently introduced GOAT-Bench [32] (see Sec. 2).

However, it diverges from the traditional ObjectNav task because, differently from the standard objective of finding any instance of a general object category, PIN requires locating a specific instance, such as "*black and white striped trekking backpack*" instead of any "*backpack*". PIN leverages zero-shot properties at the instance level, as the object instances used for the training split differ from those included in the validation episodes. This requires agents to focus on the specific characteristics of the target object defined by the input references and avoid being misled by instances of the same category that are not the actual target.

Furthermore, PIN differs from InstanceImageNav and GOAT-Bench in various aspects. First, the target object is represented by a collection of images with neutral backgrounds, rather than being shown in its current spatial context. InstanceImageNav and GOAT-Bench are based on a set of general object categories that are included in the dataset of scenes and, therefore, these objects are static and frozen in the 3D model of the environment. Instead, the peculiarity of PIN is that it is created using a set of additional photo-realistic personal objects from a collection of 3D objects that can be placed and moved in different locations of the environment between different episodes. Using additional objects allows to avoid reconstruction errors and artifacts that can distort the appearance of the target.

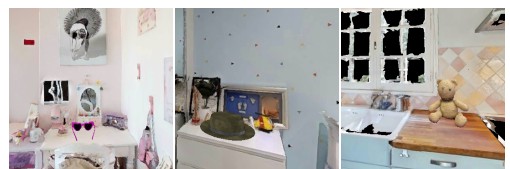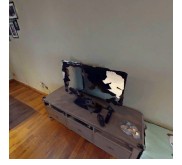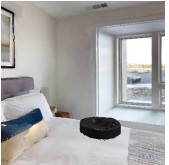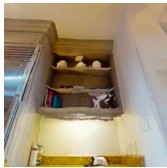

| PInNED (Ours) | InstanceImageNav | MultiON | GOAT-Bench |

Figure 2: Comparison of observations depicting different targets in the embodied setting of our PInNED dataset with the target objects of MultiON, InstanceImageNav, and GOAT-Bench datasets.

This unique characteristic compels the agent to discern and extract the defining features of the target object while maintaining invariance to the surrounding context in which it is situated since personal objects can be moved frequently and could be placed in multiple suitable locations.

Similarly to GOAT-Bench, PIN provides a multimodal input to the agent, including textual descriptions of the target instances alongside the images. However, GOAT-Bench ignores the presence of instances of the same category of the target in the scene, whereas this is the core challenge of PIN. Additionally, it is worth noting that while text alone can sometimes provide precise identification of the specific instance, it can also be ambiguous. Visual references, although generally clearer, are not always available in real-world scenarios. Therefore, the two modalities cover different real-world requirements and both deserve to be studied. An extensive comparison of current object-driven dataset properties is reported in Table 1, which presents the following columns:

- *Photo-Realistic Scenes*: the presence of photo-realistic scans taken from real-world environments (e.g. the scenes of HM3D are built from scans of real environments, while scenes in AI2-THOR are hand-built by professional 3D artists);
- *Photo-Realistic Targets*: the availability of photo-realistic objects that can be used as navigation targets. In PInNED we carefully selected objects with realistic appearances. Procedurally-generated datasets, instead, tend to favor customizability over realism;
- *Additional Objects*: the inclusion of target objects that were not present at the time of capture. Datasets like GOAT-Bench target objects which were already present in the acquired scene, while PInNED targets objects injected in the scene afterward;
- *Visual Reference*: providing visual target references for each navigation episode;
- *Descriptive Reference*: providing natural language descriptions as targets for each episode;
- *Variable Placement*: the possibility of having variable spawning positions for the targets within the dataset;
- *Instance Goal*: the inclusion of navigation episodes in which the goal is to reach the exact instance indicated to the agent.

Moreover, a qualitative comparison of goal objects observed in their position in the environment from different datasets is depicted in Fig. 2.

### 3.3 Dataset

**Categories and Instances.** We selected a pool of 18 object categories from the assets contained in Objaverse-XL dataset [22]: '*backpack*', '*bag*', '*ball*', '*book*', '*camera*', '*cellphone*', '*eyeglasses*', '*hat*', '*headphones*', '*keys*', '*laptop*', '*mug*', '*shoes*', '*teddy bear*', '*toy*', '*visor*', '*wallet*', '*watch*', for a total of 338 additional objects. Each category contains an average of 18.8 objects, with a standard deviation of 5.5. The 3D objects are selected with human supervision to ensure photo-realism and uniqueness, which are critical requirements for tackling the PIN task. Finally, the 3D models of the objects are manually rescaled to have comparable dimensions to their real-world counterparts. In this procedure, we rendered each given object in a scene from HM3D and varied the scale of the object until the result was realistic according to our judgment. Hence, each of the 338 additional objects has a manually fixed scale that is adopted when the object is injected into the navigation episodes.

**Input References.** The input images for each target personalized object are generated by rendering the 3D mesh of the object in an isolated setting. Specifically, the input images are not expected to match the camera specification of the navigating agent [36]. The digital camera undergoes a

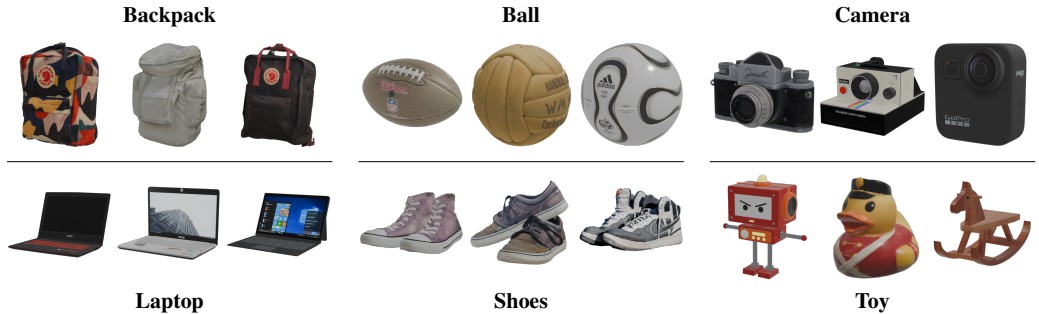

**Backpack** **Ball** **Camera**

**Laptop** **Shoes** **Toy**

Figure 3: Sample input images of personalized targets from PInNED dataset. We include three instances from various object categories within the dataset.

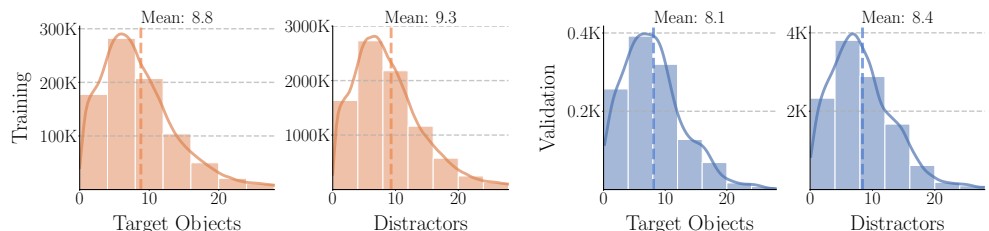

Figure 4: Plots of the distance statistics for the splits of PInNED dataset. The episodes of the training (orange) and validation splits (blue) are presented in terms of geodesic distance from the start position to the target object (left) and to all the distractors (right). All the distances are plotted in meters, and the mean value of each plot is shown on top.

30-degree yaw rotation to capture a favorable perspective of the objects. Each instance is then rotated 180 degrees in yaw to view its reverse side, followed by a 90-degree pitch rotation to observe the object from above. This procedure produces a set of three input images for each target object. An illustration of the acquired reference images is displayed in Fig. 3. Moving on to the textual references, manually annotated descriptions are produced for each target personalized object with the scope of highlighting the details that allow the agent to distinguish it from other instances of the same category. Specifically, we provide three descriptions for each personalized object in the PInNED dataset. To annotate the descriptions, we provided two object instances at a time to the annotators, asking them to describe one of the two objects in such a way that it is distinguishable from the other. This procedure results in a total of 960 unique words and an average of 10.7 words per description. Additional samples of input references are included in the Appendix.

**Scenes.** The benchmark defined by the PIN dataset is situated in the indoor photo-realistic scenes (*e.g.*, apartments, offices, houses) within the semantically-annotated subset [74] of Habitat-Matterport3D (HM3D) [56] which consists of 145 environments for the training split and 36 for validation set. However, one validation scene is ignored as it represents an art gallery and has no suitable spawnable surfaces. HM3D was selected due to its status as the largest publicly available dataset of semantically annotated indoor spaces with photo-realistic quality for embodied navigation.

**Episode Generation.** During the generation of the dataset, the bounding boxes of the surfaces in the environment are extracted using the semantic annotations of the scene. To obtain the bounding box from the texture, we extracted the point cloud 3D model of each scene and ensured that each point retained its associated annotation color. Subsequently, points were clustered by annotation color to create the bounding box associated with each piece of furniture. The spawning position of each object is selected by sampling from the positions of a curated set of suitable surface macro-categories included in the semantic annotations of HM3D. The surface categories selected for the creation of the dataset are: *armchair, bed, bench, cabinet, piano, rug, sofa, table*. These specific surfaces are chosen because of the high probability of personalized objects being positioned on top.

In each episode of the PInNED dataset, a single instance of a specific category is chosen as the target object. Consequently, up to 6 instances belonging to the same category, and up to 13 objects from other categories, are added to the environment as distractors. All additional objects placed in

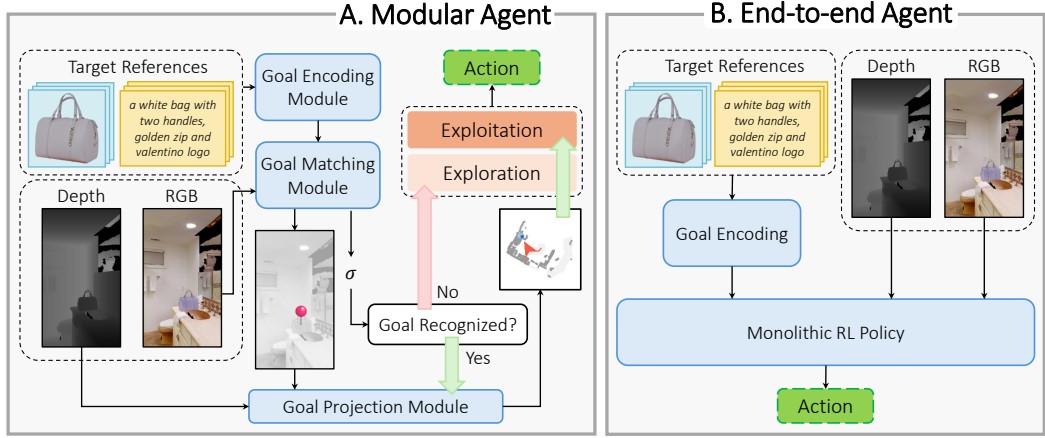

Figure 5: Overview of the baselines designed for the PIN task: modular agent (on the left) and end-to-end agent based on a monolithic reinforcement learning-based policy (on the right).

the environment are constrained to be on the same level/floor as the agent by selecting spawnable surfaces with a bounding box position within $0.5$ meters from the starting position of the agent along the vertical axis. For each environment in the training split a set of 400 episodes is sampled for each one of the possible categories. For the generation of the validation split each target category is used twice. Finally, episodes where the target object is not reachable by an agent following the shortest path are removed from the dataset. Refer to the Appendix for more details on dataset generation.

The resulting dataset for PIN is defined by a total of $865,519$ generated episodes for the training split, while the validation split contains $1,193$ episodes. The geodesic distances of the target and distractors from the starting position of the agent in the episodes of PInNED are shown in Fig. 4. In the figure, the distribution of the distances of targets and distractors significantly overlap, hence prior information on the target object distance is hardly exploitable.

## 4 Baselines

In this section, we present the set of approaches that are revisited and tested on our introduced PInNED dataset. These methods are recent object-driven methods and can be grouped into two categories: (i) **modular agents** that decouple the navigation task into specialized sub-modules and (ii) **end-to-end agents** based on a monolithic policy trained using reinforcement learning. Fig. 5 shows an overview of these two approaches. We refer to the Appendix for more details on the implementation of the baselines.

### 4.1 Modular Agents

In recent years, modular agents gathered an increasing interest in various embodied settings. These agents tackle the high-level navigation tasks by decoupling them into a chain of specialized sub-modules, each of which handles a smaller task. Specifically, Chaplot *et al.* [15] proposed SemExp, a modular agent designed for the ObjectNav task composed of three main modules: exploration, object detection, and exploitation. The core idea is that the agent explores as much as possible the unseen environment while the detection module localizes the semantic objects in the acquired observations. Inspired by this approach, Mod-IIN [35] and CLIP on Wheels (CoW) [29] adapt the detection module to handle specific instances and open-vocabulary targets, respectively. For our modular agent baselines, we consider the same exploration and exploitation modules used in these previous works, while changing and adapting the object detection module for the PIN task.

**Exploration Module.** The exploration module is entitled to explore the unseen areas of the environment with the scope of encountering the target object. As in Mod-IIN and CoW, we adopt a frontier-based exploration (FBE [75]) approach. The agent builds an occupancy map of the environment during navigation, and at every time step, if the goal is not detected, the unexplored frontier on the map which is closest to the agent is selected as the current goal.

**Object Detection Module.** The object detection module receives the visual or textual references and the current RGB observation of the agent. Then, it is tasked with providing (i) a **matching score** that, whether it exceeds a certain matching threshold $\sigma$, determines that the goal has been recognized; and (ii) a **series of coordinates** on the observation which correspond to where the goal is located, that are used by the exploitation module to project the goal on a 2D map. We select three categories of approaches to implement this module:

- 📌 **Keypoint Matching**: In this category, the visual target references and the current RGB observation are provided to a keypoint matching method. We tested SuperGlue [60], following the approach proposed by Mod-IIN [35], and the framework introduced in IEVE [40]. In particular, SuperGlue outputs a confidence score for each matched keypoint pair. We use the sum of these confidences as the matching score and the keypoints that exceed a given confidence threshold $\tau$ as the localization coordinates. Regarding the Exploration-Verification-Exploitation framework proposed in IEVE, we adapted some components to match the different requirements of our task. Specifically, we first collected an auxiliary dataset, which includes, for each goal in the training set, 10 positive samples and one negative sample containing a distractor from the same category as the goal. We trained InternImage [69] to classify the 18 categories of our dataset using the goal images of the training set. Instead of the InternImage segmentation model, since, to the best of our knowledge, no segmentation dataset contains all our categories, we adopted the open-vocabulary segmenter GroundedSAM [59]. For the image-matching step, we exploited LightGlue [42] on the keypoints extracted with DISK [67] as in the original IEVE paper.

- 📌 **Patch-level Matching**: A Vision Transformer (ViT [26]) encoder divides an image into patches and extracts patch-level embeddings. Hence, we extract a goal embedding from each reference and compute the cosine similarity with the patch-level feature vectors of the RGB observation. If at least a patch has a similarity that exceeds the matching threshold $\sigma$, the goal is considered detected. The center coordinates of these patches are used as the goal localization result. For the visual references, we employ DINO [12], DINOv2 [51], and CLIP [54] performing a region pooling over the reference objects to obtain goal feature vectors. For the textual references, a text-aligned multimodal encoder is required. Hence, we employ CLIP and, inspired by [29], CLIP with gradient relevance [17] (CLIP-Grad). We assume the mean embedding of the set of prompt templates used in CoW applied to the target descriptions as the target feature vector.

- 📌 **Detection Model**: We consider detection models that produce output regions according to a given reference. Specifically, we consider PerSAM [80] (both in the standard and one-shot finetuned versions) and OWL [49], which localize regions according to, respectively, visual and textual references. As in CoW, we exploit the output confidence to determine whether the goal has been detected and return the central coordinates of the region as the goal localization result.

**Exploitation Module.** The exploitation module takes control of the navigation when the goal is recognized in the current observation. After detecting the target object at a given location, the exploitation module is triggered and computes the route to reach the target object. The goal position provided by the object detection module is projected into an occupancy map, and the Fast Marching Method [16, 62] is used to plan the path from the current position of the agent to the detected goal position. When the agent reaches the goal position, the '*stop*' action is called to conclude the episode.

## 4.2 End-to-End Agents

In contrast to modular agents, end-to-end approaches train a neural network policy to process sensor input and predict the atomic actions needed to complete the required task. We consider two recent approaches for embodied navigation and adapt them for the Personalized Instance-based Navigation task: (i) ZSON [45], which pre-trains an ImageNav agent and evaluates downstream on ObjectNav leveraging the capabilities of CLIP multimodal embeddings; and (ii) RIM [18], which employs a Transformer-based architecture [68] that is trained using auxiliary tasks and uses a recursive implicit map that is updated during the navigation for the ObjectNav task. We finetune both approaches on PInNED dataset. Specifically, ZSON is adapted to use image references as input during its ImageNav pretraining phase. While, for RIM, we employ two finetuning strategies: conditioning the navigation on textual features extracted from the reference descriptions and conditioning on visual features extracted from the image references. The features produced using both modalities of PInNED references are extracted using CLIP.

Table 2: Navigation results on PInNED on the environments of HM3D dataset, considering the presence of distractors from the same category. **Bold** text denotes the best performance among each category of approaches.

| | Backbone | Modality | Navigation Metrics | | | | | Detection Metrics | | | |
|---|---|---|---|---|---|---|---|---|---|---|---|
| | | | SR↑ | SPL↑ | CE↓ | D2G↓ | Steps | %Match↑ | TM↑ | CM↓ | NM↓ |
| *Modular Agents* | | | | | | | | | | | |
| CLIP [54] | ViT-B/16 | Textual | 3.10 | 1.82 | 9.31 | 7.94 | 503.1 | 62.95 | 20.07 | 22.07 | 57.86 |
| CLIP-Grad [29] | ViT-B/32 | Textual | 4.53 | 2.42 | 6.95 | 7.94 | 465.8 | 77.95 | 4.65 | 7.21 | 84.14 |
| OWL [29, 49] | ViT-B/32 | Textual | 7.29 | 3.36 | 12.66 | 7.90 | 871.7 | 22.97 | **62.60** | 32.88 | 4.52 |
| SuperGlue [35, 60] | - | Visual | 3.27 | 1.28 | 7.38 | 8.36 | 804.0 | 29.42 | 16.96 | 3.44 | 79.60 |
| IEVE [40] | - | Visual | 3.52 | 3.07 | 12.25 | 7.73 | 712.1 | 30.03 | 32.39 | 16.01 | 51.60 |
| PerSAM [80] | ViT-B/16 | Visual | 2.77 | 1.81 | 6.53 | 8.20 | 362.5 | **81.98** | 1.15 | 10.43 | 88.42 |
| PerSAM-F [80] | ViT-B/16 | Visual | 1.93 | 1.28 | **5.63** | 8.12 | 321.3 | 36.13 | 0.60 | 13.48 | 85.92 |
| DINO [12] | ViT-B/16 | Visual | 4.02 | 1.71 | 6.88 | 8.28 | 826.0 | 23.89 | 62.73 | **1.36** | 35.91 |
| CLIP [54] | ViT-B/16 | Visual | 9.64 | 5.39 | 13.33 | 7.79 | 623.5 | 58.51 | 32.53 | 16.35 | 51.12 |
| DINOv2 [51] | ViT-B/14 | Visual | **14.84** | **7.94** | 26.15 | **7.28** | 658.7 | 55.74 | 55.33 | 42.00 | **2.67** |
| *End-to-end Agents* | | | | | | | | | | | |
| RIM [18] | ResNet-50 | Textual | 7.12 | 6.67 | **10.44** | 8.43 | 409.3 | - | - | - | - |
| RIM [18] | ResNet-50 | Visual | 8.80 | 6.80 | 13.41 | 8.48 | 402.1 | - | - | - | - |
| ZSON [45] | ResNet-50 | Visual | 9.14 | 7.18 | 21.12 | **7.00** | 389.9 | - | - | - | - |

## 5 Experimental Evaluation

In this section, we present an experimental analysis of the selected baselines on the PIN task, discussing the set of metrics used to effectively evaluate the performances and the obtained results.

### 5.1 Evaluation Metrics

Traditional metrics for object-driven embodied navigation are **success rate** (SR) and **success rate weighted by path length** (SPL). SR is the ratio between the number of episodes where the agent successfully reaches the target object within a maximum distance of 1 meter and the total number of episodes, while SPL weighs the success rate with the length of the path taken by the agent. Moreover, we report the **average number of steps** taken by the agent and the **average distance from the goal** (D2G) at the end of each episode. The agent designed for tackling the PIN task should be able to distinguish whether the target object is present in the current observation while exploring the unseen environment and correctly localize it, within the timesteps budget $T$ (set to 1,000). The main challenge is represented by distractor instances belonging to the same category as the target object. Hence, we introduce the **category error** (CE) metric, which measures the percentage of episodes in which the agent stopped within one meter from instances belonging to the same category of the goal.

In modular agents, the ability to detect the correct instance resides in having large matching scores when the target is present in the observation and small scores when the target is absent. Since in these agents it is possible to determine whether a given observation matches, we compute four additional metrics: the **percentage of episodes with at least a detected match** (%Match), the **percentage of matched observations** that contain the **target object** (TM), an **instance of the same category of the target** (CM), or **no relevant objects** (NM).

### 5.2 Experimental Results

**Personalized Instance-based Navigation Experiments.** In Table 2, we present the results on the PIN task. Among modular agents, DINOv2 performs best according to SR and SPL. The high values of TM, CM, and CE show that the obtained matches usually refer to the same category of the target instance. The same reasoning can be applied to OWL for the modular agents using textual references. However, OWL produces fewer matches as can be noted from the %Match metric. Models such as SuperGlue, PerSAM, and PerSAM-F, which exhibit low SR and TM, have also a corresponding high NM, demonstrating that they are not able to provide significant matching scores for distinguishing the correct instances or even the correct categories. It is noteworthy that SuperGlue struggles to match the instances of PInNED, which are represented on a neutral background, contrary to InstanceImageNav [35], where the reference image is a photo of the object

Table 3: Navigation results on PInNED on the environments of HM3D dataset, without considering the presence of distractors from the same category of the target. **Bold** text denotes the best performance among each category of approaches.

| | Backbone | Modality | Navigation Metrics | | | | Detection Metrics | | |
|---|---|---|---|---|---|---|---|---|---|
| | | | SR↑ | SPL↑ | D2G↓ | Steps | %Match↑ | TM↑ | NM↓ |
| *Modular Agents* | | | | | | | | | |
| CLIP [54] | ViT-B/16 | Textual | 3.35 | 1.86 | 8.01 | 516.5 | **61.86** | 22.83 | 77.17 |
| OWL [29, 49] | ViT-B/32 | Textual | 8.22 | 3.18 | 7.88 | 929.9 | 13.83 | 93.91 | 6.09 |
| CLIP [54] | ViT-B/16 | Visual | 11.15 | 5.92 | 7.65 | 666.2 | 52.56 | 35.57 | 64.43 |
| DINOv2 [51] | ViT-B/14 | Visual | **23.13** | **11.61** | **6.62** | 784.5 | 38.64 | **96.09** | **3.91** |
| *End-to-end Agents* | | | | | | | | | |
| RIM [18] | ResNet-50 | Textual | 7.46 | 6.87 | 7.94 | 487.1 | - | - | - |
| RIM [18] | ResNet-50 | Visual | 10.35 | 7.53 | 7.75 | 475.9 | - | - | - |
| ZSON [45] | ResNet-50 | Visual | **10.39** | **8.00** | **6.91** | 460.1 | - | - | - |

in the same context in which it is located. Regarding PerSAM and PerSAM-F, the results show that the feature space of SAM [33] is not informative enough to understand whether an instance is present in an observation. IEVE shows an improvement with respect to the other image-matching modular agent, based on SuperGlue. This is motivated by the fact that IEVE, differently from other image-matching approaches, combines LightGlue with a semantic detector, allowing the agent to focus only on observations that contain objects of the target category. This behavior is confirmed by the increased numbers of target matches, category matches, and category errors.

Moreover, end-to-end agents tend to perform worse than modular agents. This can be attributed to the imitation training performed using the ground-truth trajectory to the goal. Since in the PIN task the target instances can be placed in multiple locations, it is not possible to exploit prior semantic knowledge about the estimated location of the target instance. Moreover, end-to-end agents tend to struggle in backtracking and in recovering the navigation when moving in the wrong direction. This behavior can also be noted from the path length, which for end-to-end agents is shorter than modular agents, that continue the exploration until the whole environment is observed.

**Ablation on Category Distractors.** In Table 3, we introduce an ablation study in which we remove the distractors belonging to the same category of the target instance. Overall, metrics for all the agents improve because the presence of these distractors represents the core challenge of the PIN task. In particular, DINOv2 improves by 8.29 with respect to the main experiments, demonstrating that it embeds strong semantic correspondence properties among the same category, but that it is not trivial to identify a threshold that clearly distinguishes specific instances. The impact of same-category distractors on end-to-end agents is minor since they are finetuned to identify the correct instance.

# 6 Conclusion

In this work, we presented the task of Personalized Instance-based Navigation (PIN) in which the agent is required to locate and navigate toward a specific target instance. Additionally, we release PInNED, a task-specific dataset built by injecting a set of additional photo-realistic objects in the scenes of HM3D. Finally, we perform an extensive analysis of recent navigation methods adapted for the proposed task. Experimental results demonstrate that the new challenges in the recognition of specific instances introduced in our proposed task are still far from being addressed. This benchmark sets a novel testbed for future work on embodied navigation toward personalized instances.

## Acknowledgments and Disclosure of Funding

This work has been conducted under a research grant co-funded by Leonardo S.p.A. and supported by the EU Horizon project "ELIAS - European Lighthouse of AI for Sustainability" (No. 101120237), the project "Personalized Robotics as Service Oriented Applications (PERSEO)" funded under the Marie Sklodowska-Curie Action Horizon 2020 (No. 955778), and the PNRR project "Fit for Medical Robotics (Fit4MedRob)" funded by the Italian Ministry of University and Research.

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
