# 📌 Personalized Instance-based Navigation Toward User-Specific Objects in Realistic Environments

## Supplemental Material

**Luca Barsellotti**[*]    **Roberto Bigazzi**[*]
**Marcella Cornia    Lorenzo Baraldi    Rita Cucchiara**
University of Modena and Reggio Emilia, Italy
`{firstname.lastname}@unimore.it`

Project page: aimagelab.github.io/pin

## A    Dataset and Codebase Release

The dataset and codebase of our work are released at the following link[1]. We provide the instructions to download the assets contained in the PInNED dataset and the codebase to run the main experiments on the Personalized Instance-based Navigation (PIN) task.

## B    Limitations

A limitation of this work is related to the visual appearance of some of the object instances in the PInNED dataset. For example, the Habitat simulator's [61] rendering can cause a deterioration in the texture quality of some objects, failing to accurately reproduce them in the environment. Moreover, instances with very small or detailed components can also exhibit a degradation in their visual fidelity when instantiated in the simulator. Consequently, as the agent moves farther from these objects, their details become less discernible. As a direct consequence, detecting small target objects is a critical challenge for navigation agents tackling the PIN task.

This behavior is showcased in Sec. E, where agents tackling the PIN task in the episodes of PInNED dataset face significant challenges in successfully detecting instances of inherently small object categories. In fact, despite agents such as the modular agent with DINOv2 [51] showcase good performance on the overall PIN task, detecting small objects represents one of the main limitations of current object-driven agents, as they can only be recognized when the robot is close to them.

A possible future improvement could involve designing novel exploration policies that aim to bring the robot closer to surfaces where the target might be placed while leveraging different detection criteria that take into consideration the scale of the observed objects.

## C    Broader Impacts

The introduction of the Personalized Instance-based Navigation (PIN) task and the accompanying PInNED dataset has the potential to advance the field of visual navigation and Embodied AI. The PIN task fills the limitations of the current datasets for embodied navigation by requiring agents to distinguish between multiple instances of objects from the same category, thereby enhancing their precision and robustness in real-world scenarios. This advancement can lead to more capable and reliable robotic assistants and autonomous systems, especially in household settings. Moreover, the PInNED dataset serves as a comprehensive benchmark for the development and evaluation of

---

[*]Equal contribution.
[1]https://github.com/aimagelab/pin

38th Conference on Neural Information Processing Systems (NeurIPS 2024) Track on Datasets and Benchmarks.

Table A: Configuration of the main parameters used for executing each episode of the PIN task contained in the PInNED dataset.

| Action Space | | Episode Configuration | | Depth Sensor | |
|---|---|---|---|---|---|
| forward step | 0.25 | success distance | 1.0 | width | 360 |
| turn angle | 30 | max episode steps | 1000 | height | 640 |
| tilt angle | 30 | **RGB Sensor** | | hfov | 42 |
| **Agent Configuration** | | width | 360 | position | [0, 1.31, 0] |
| visual sensors | rgb, depth | height | 640 | min depth | 0.5 |
| height | 1.41 | hfov | 42 | max depth | 5.0 |
| radius | 0.17 | | | | |
| position | [0, 1.31, 0] | | | | |

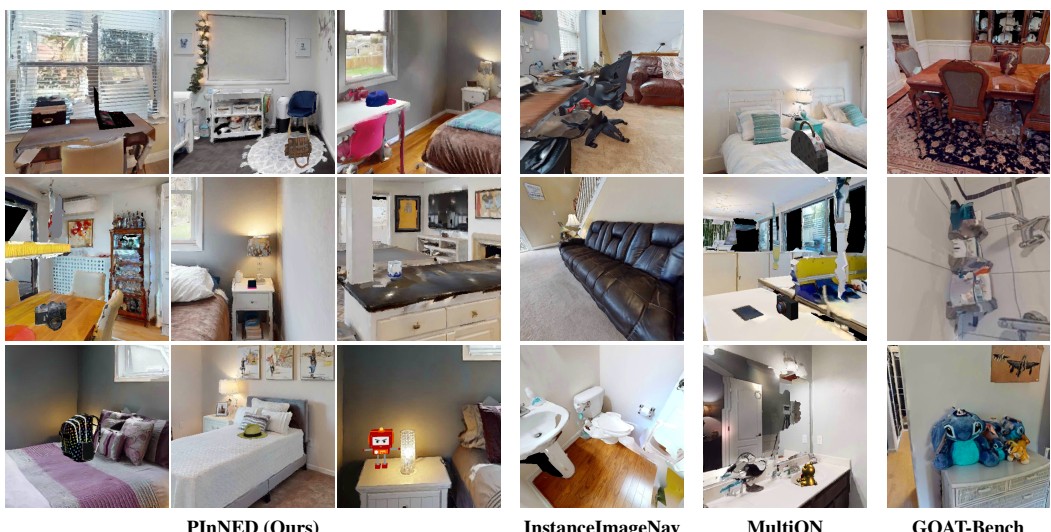

**PInNED (Ours)**  **InstanceImageNav**  **MultiON**  **GOAT-Bench**

Figure A: Comparison of observations depicting different target objects of PInNED dataset with the target objects of InstanceImageNav, MultiON, and GOAT-Bench datasets.

novel algorithms in object-driven navigation. By providing a challenging and extensive dataset, we encourage the research community to develop innovative approaches and solutions.

## D   Additional Personalized Instance-based Navigation Details

**Configurations.** In addition to the task definition details provided in Sec. 3.1 of the main paper, relevant hyperparameters employed for executing each episode of the PInNED dataset are presented in Table A.

The configuration used for a PIN episode comprises a maximum duration of $1,000$ time steps, with the agent's action space defined by discrete forward steps of $0.25$ m, a turn angle of $30°$, and a head tilt angle of $30°$. Each episode is considered successful if the position of the agent is within 1 meter from the position of the target object, and it predicts the '*stop*' action before the end of the time step budget. The configurations used for the navigation experiments reflect the settings employed to simulate the camera sensors and space occupation of the HelloRobot Stretch[2] platform.

**Comparison with Object-oriented Tasks.** In addition to Fig. 2 of the main paper, in Fig. A we showcase additional examples of goal objects captured in the embodied setting for different object-driven datasets. The target objects belonging to the PInNED dataset are compared with InstanceImageNav [36], MultiON [70], and GOAT-Bench [32] datasets. It is noticeable that injecting photo-realistic objects allows to have targets that do not present artifacts or reconstruction errors, which is common for InstanceImageNav and GOAT-Bench target objects. Furthermore, when comparing the target objects of PInNED with those in the MultiON dataset, it is noticeable that the PInNED objects exhibit a more photo-realistic visual quality.

---

[2]https://hello-robot.com/stretch

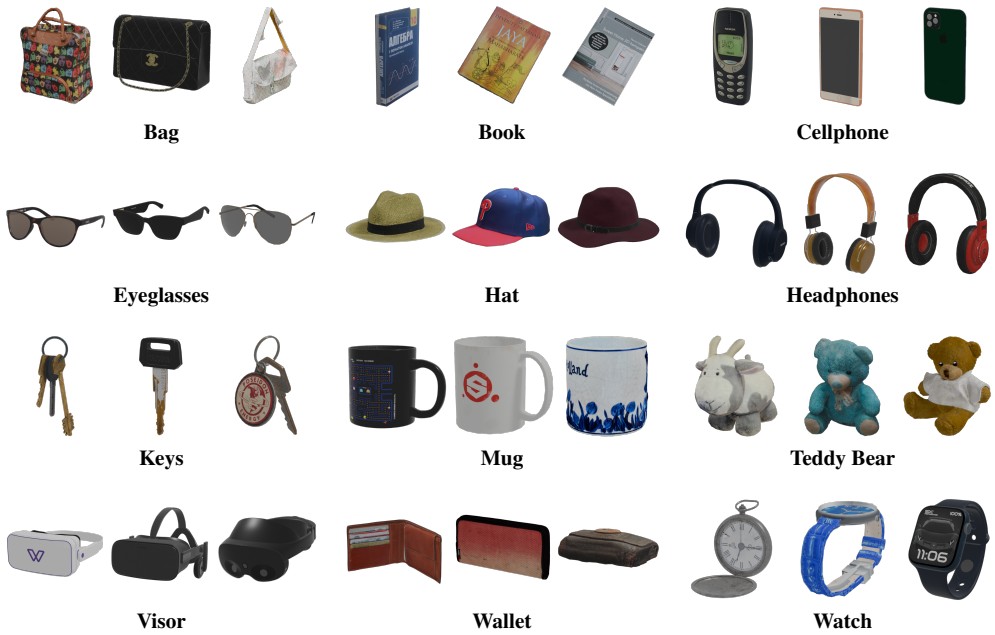

**Figure B:** Sample frontal visual references of personalized targets from PInNED dataset. We include three instances for each object category, considering the categories not included in Fig. 3 of the main paper.

**Comparison with ProcTHOR.** ProcTHOR [23] is a framework built on AI2-THOR [34] to procedurally generate interactive environments, enabling the evaluation of data augmentation and large-scale training in different Embodied AI tasks. PInNED is a dataset designed specifically to study the newly introduced PIN task, in which the agent is tasked with finding a specific instance according to target images or textual descriptions.

ProcTHOR includes 1,633 instances across 108 object categories, with the ability to vary brightness, colors, materials, and object states. These categories include several household objects, covering generic objects, such as '*pen*' or '*apple*', objects that can be personal, such as '*mug*' and '*watch*', and large objects that are unlikely to change their placement in the environment, such as '*fridge*', and '*window*'. PInNED presents 18 object categories that can be personal, with the specific purpose of accompanying the task in which the agent has to distinguish instances belonging to the same category. All the categories represent objects that can be moved frequently in the environment and do not have a predefined location.

As well as most procedural datasets, ProcTHOR sacrifices realism in favor of interactivity, scalability, and customizability. PInNED, as a task-specific dataset, favors photo-realistic environments and objects. Indeed, it is the first instance-based navigation dataset based on both photo-realistic environments and injected objects, that can be moved frequently and with multimodal targets. Interactivity with the objects is out of scope for this work, however, the addition of external objects paves the way for possible future enhancements where interactivity is needed.

# E  Additional PInNED Dataset Details

**Additional Reference Samples.** To better visualize the content of PInNED dataset, in Fig. B we illustrate additional samples of the acquired visual references for the categories that are not included in Fig. 3 of the main paper.

Additionally, we present samples including both visual and textual modalities for the input references associated with some of the object instances of PInNED dataset in Fig. C and Fig. D. In particular, we show the three views composing the set of visual references and the three manually annotated descriptions for the textual references.

Table B: Statistics about the number of distractors placed in the episodes of the training and validation sets of PInNED dataset. We consider the distractors belonging both to the same category of the target and to other categories.

| # of Distractors | Same Object Category | | Other Categories | |
|---|---|---|---|---|
| | **Train** | **Val** | **Train** | **Val** |
| Max | 6 | 3 | 13 | 10 |
| Average | 2.93 | 2.90 | 7.75 | 7.19 |
| Standard Deviation | 0.33 | 0.37 | 2.84 | 2.82 |

**Object Selection and Distribution Criteria.** The scope of PIN is to provide a benchmark to evaluate an agent tasked with finding a specific object that can be located anywhere in an unexplored environment, where distractors of the same category are present; hence, the object categories are selected according to the following criteria: (i) objects that are highly customizable in terms of shapes, colors, and other visual aspects, (ii) objects that are frequently moved and can be placed anywhere, and (iii) objects of common use for which is reasonable to ask a robot to find.

**Additional Information about Dataset Generation.** In Table B, we provide statistics on the number of distractors placed in the training and validation episodes of PInNED dataset. During the generation of PIN episodes, a maximum number of distractors, both from the same category as the target instance and from other categories, is sampled from the set of available objects. The final number of additional objects in each episode is determined by the number of suitable surfaces and the available space on these surfaces. During the dataset generation process, objects are positioned above these surfaces and lowered until they contact the surface. If an object cannot be initially placed due to size constraints or collisions with other elements or walls, the placing process for that object is aborted, and another one is sampled from unused object instances. After the generation of the dataset of episodes, an additional assessment is performed through the Habitat simulator to remove the episodes containing objects that are not reachable from the starting position of the agent.

**Object Distances.** In addition to Fig. 4 of the main paper, Fig. E presents a plot depicting the Euclidean distances of target objects and distractors from the starting position of the agent in the episodes of both training and validation splits of PInNED dataset. When considering the Euclidean distance, the distribution of the distances of the additional objects remains consistent with the geodesic distances presented in the main paper. Furthermore, the plots of the distances of all additional objects (target instances and distractors) are presented in Fig. F.

**Modular Agent Activations.** In Fig. G we present a comparison of the similarities computed between the patch-level features of different backbones on the observations of the agent and the references. In particular, we show these similarities on DINOv2 [51], DINO [12], CLIP with visual references, and CLIP with textual references [54]. The resolution of the similarities extracted from DINOv2 is higher than the others since we employed the input resolution $518 \times 518$ on which the ViT-B/14 model has been trained, which corresponds to a grid of $37 \times 37$ patches, whereas DINO and CLIP are based on a ViT-B/16 backbone with $224 \times 224$ as input resolution. It is noteworthy that DINOv2 exhibits strong semantic localization properties, with high similarity values on the exact location of the image on which the target is observed. On the contrary, DINO and CLIP tend to exhibit less well-localized similarities. Moreover, CLIP with visual references has a high similarity on the patches corresponding to the laptop in the observation, whereas CLIP with textual references has a low similarity on the same patches.

**Object Size Analysis.** Taking into account that personalized objects are defined as predefined instances with distinct characteristics, the primary challenge in the PIN task lies in effectively recognizing these specific details, especially when dealing with subtle features and limited interaction capabilities within the environment. In this analysis, we present a category-wise size analysis of the objects in the dataset by computing and measuring the 3D bounding box of each object. In Fig. H, we plot the distribution of the volumes of the bounding boxes associated with each object category showing that the distributions between training and validation splits remain consistent.

**Category-wise Navigation Results.** In Table C we present the navigation results of the modular agent based on DINOv2 as the matching backbone in which we compute the metrics for each category. From the results on SR and SPL we can note that there are categories that are easier to locate and reach, such as '*backpack*', '*bag*', '*ball*', '*hat*', '*laptop*', and '*toy*', and there are instances from

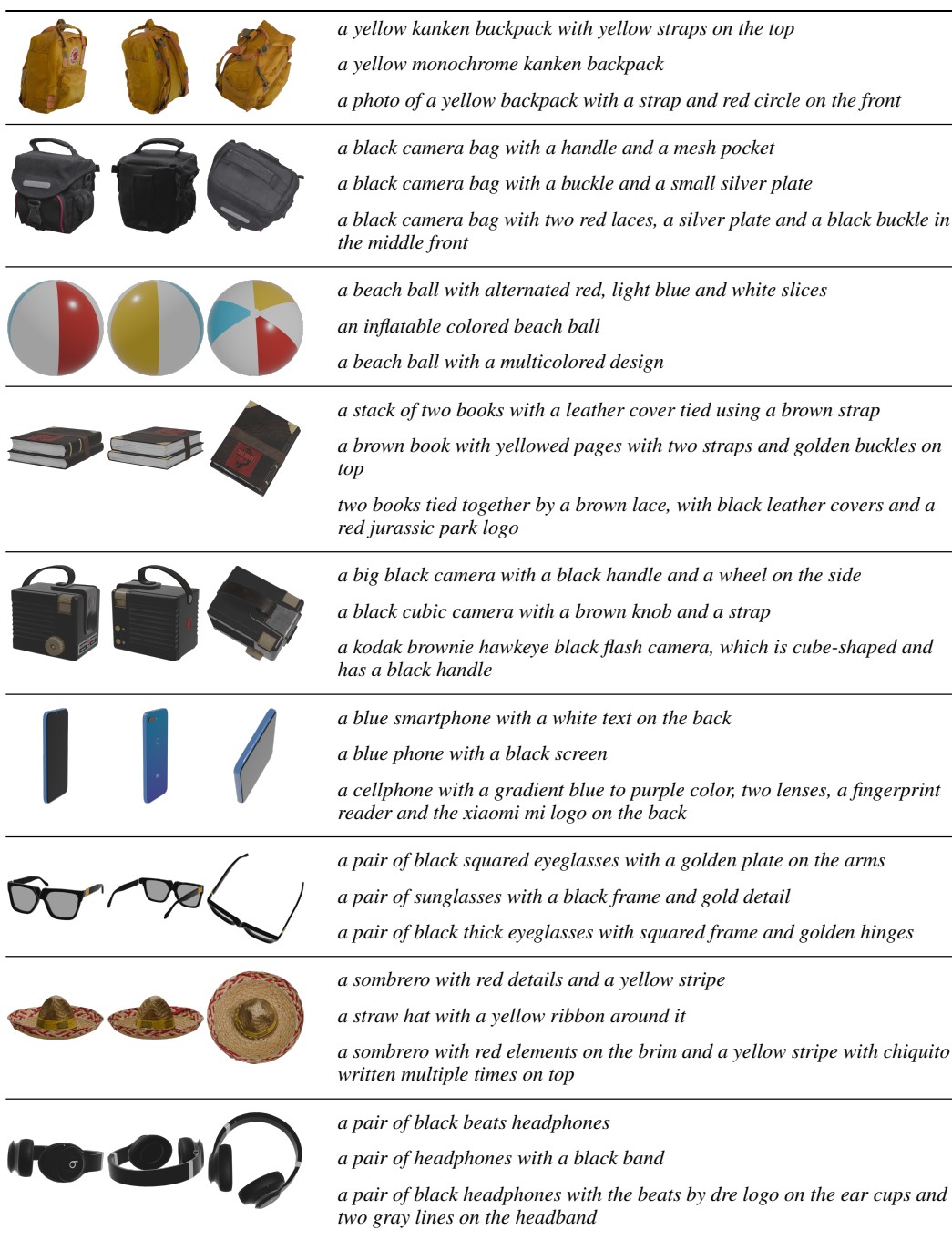

| | |
|---|---|
| | *a yellow kanken backpack with yellow straps on the top* |
| | *a yellow monochrome kanken backpack* |
| | *a photo of a yellow backpack with a strap and red circle on the front* |
| | *a black camera bag with a handle and a mesh pocket* |
| | *a black camera bag with a buckle and a small silver plate* |
| | *a black camera bag with two red laces, a silver plate and a black buckle in the middle front* |
| | *a beach ball with alternated red, light blue and white slices* |
| | *an inflatable colored beach ball* |
| | *a beach ball with a multicolored design* |
| | *a stack of two books with a leather cover tied using a brown strap* |
| | *a brown book with yellowed pages with two straps and golden buckles on top* |
| | *two books tied together by a brown lace, with black leather covers and a red jurassic park logo* |
| | *a big black camera with a black handle and a wheel on the side* |
| | *a black cubic camera with a brown knob and a strap* |
| | *a kodak brownie hawkeye black flash camera, which is cube-shaped and has a black handle* |
| | *a blue smartphone with a white text on the back* |
| | *a blue phone with a black screen* |
| | *a cellphone with a gradient blue to purple color, two lenses, a fingerprint reader and the xiaomi mi logo on the back* |
| | *a pair of black squared eyeglasses with a golden plate on the arms* |
| | *a pair of sunglasses with a black frame and gold detail* |
| | *a pair of black thick eyeglasses with squared frame and golden hinges* |
| | *a sombrero with red details and a yellow stripe* |
| | *a straw hat with a yellow ribbon around it* |
| | *a sombrero with red elements on the brim and a yellow stripe with chiquito written multiple times on top* |
| | *a pair of black beats headphones* |
| | *a pair of headphones with a black band* |
| | *a pair of black headphones with the beats by dre logo on the ear cups and two gray lines on the headband* |

Figure C: Visual reference images and textual reference descriptions of personalized targets from PInNED dataset. The samples are taken from '*backpack*', '*bag*', '*ball*', '*book*', '*camera*', '*cellphone*', '*eyeglasses*', '*hat*', and '*headphones*' object categories.

categories that are never correctly reached, such as '*keys*', '*wallet*', and '*watch*'. This result returns the inability of the vanilla matching modules to distinguish these categories in the embodied setting. Moreover, we can observe that there is an overall positive correlation between SR and average category size, implying that small objects are particularly challenging to detect.

**Similarity Analysis.** The similarity of objects is a critical factor in the PIN task. The presence of distractors increases the challenge of the proposed task, as the agent must balance between being overly cautious and overly confident when identifying target instances. This trade-off is central to

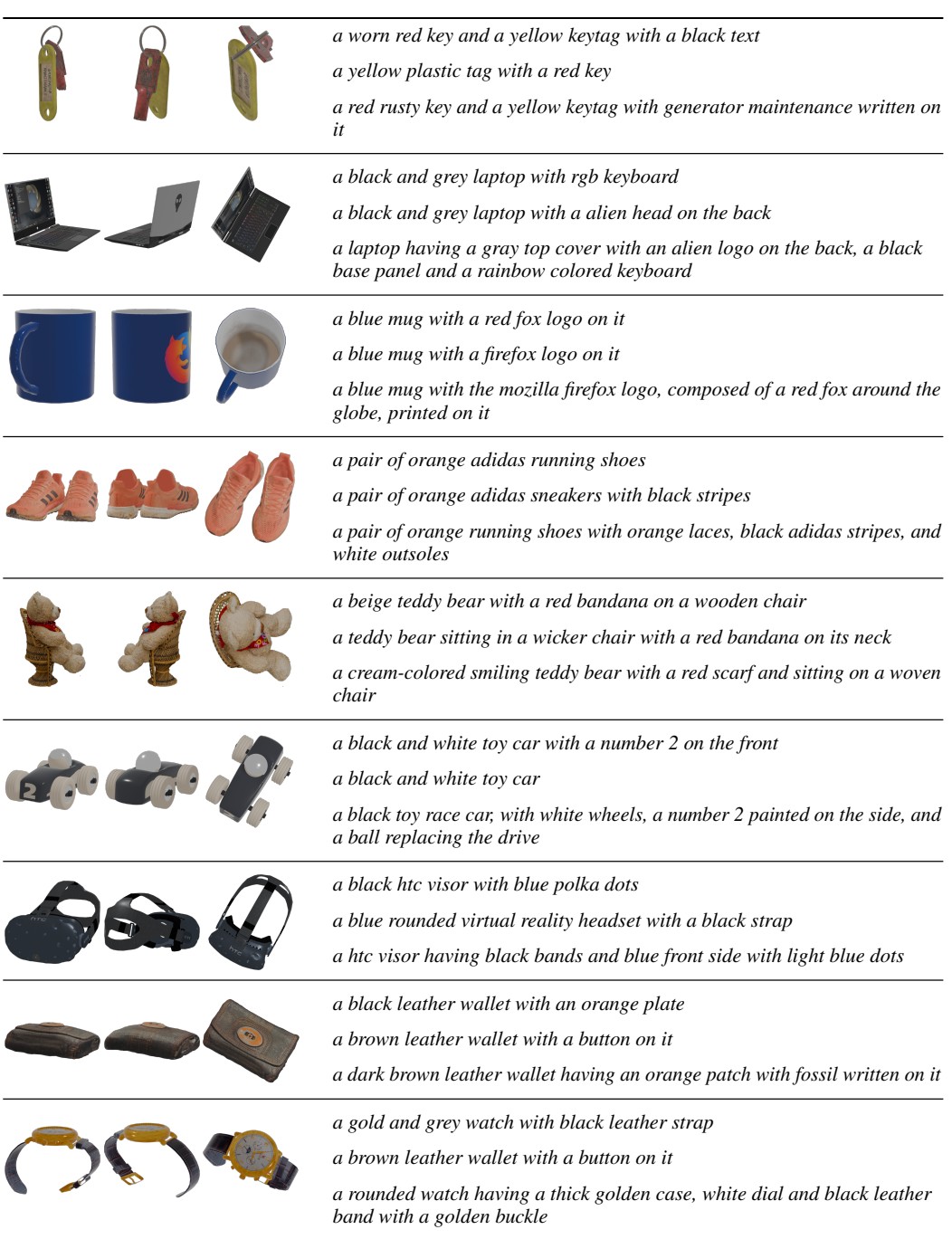

| | |
|---|---|
| | *a worn red key and a yellow keytag with a black text* |
| | *a yellow plastic tag with a red key* |
| | *a red rusty key and a yellow keytag with generator maintenance written on it* |
| | *a black and grey laptop with rgb keyboard* |
| | *a black and grey laptop with a alien head on the back* |
| | *a laptop having a gray top cover with an alien logo on the back, a black base panel and a rainbow colored keyboard* |
| | *a blue mug with a red fox logo on it* |
| | *a blue mug with a firefox logo on it* |
| | *a blue mug with the mozilla firefox logo, composed of a red fox around the globe, printed on it* |
| | *a pair of orange adidas running shoes* |
| | *a pair of orange adidas sneakers with black stripes* |
| | *a pair of orange running shoes with orange laces, black adidas stripes, and white outsoles* |
| | *a beige teddy bear with a red bandana on a wooden chair* |
| | *a teddy bear sitting in a wicker chair with a red bandana on its neck* |
| | *a cream-colored smiling teddy bear with a red scarf and sitting on a woven chair* |
| | *a black and white toy car with a number 2 on the front* |
| | *a black and white toy car* |
| | *a black toy race car, with white wheels, a number 2 painted on the side, and a ball replacing the drive* |
| | *a black htc visor with blue polka dots* |
| | *a blue rounded virtual reality headset with a black strap* |
| | *a htc visor having black bands and blue front side with light blue dots* |
| | *a black leather wallet with an orange plate* |
| | *a brown leather wallet with a button on it* |
| | *a dark brown leather wallet having an orange patch with fossil written on it* |
| | *a gold and grey watch with black leather strap* |
| | *a brown leather wallet with a button on it* |
| | *a rounded watch having a thick golden case, white dial and black leather band with a golden buckle* |

Figure D: Visual reference images and textual reference descriptions of personalized targets from PInNED dataset. The samples are taken from '*keys*', '*laptop*', '*mug*', '*shoes*', '*teddy bear*', '*toy*', '*visor*', '*wallet*', and '*watch*' object categories.

the effectiveness of the navigation approaches. In particular, concerning images as references of the target object, re-identification methods should be a robust solution against distractors due to considering the matching between keypoints instead of the semantic similarity between observation and reference. Indeed, in Table 2 of the main paper, the state-of-the-art re-identification method SuperGlue has a lower category error than DINOv2 and CLIP. However, it presents the worst results according to SR and SPL, showing difficulties in matching keypoints when observation and reference

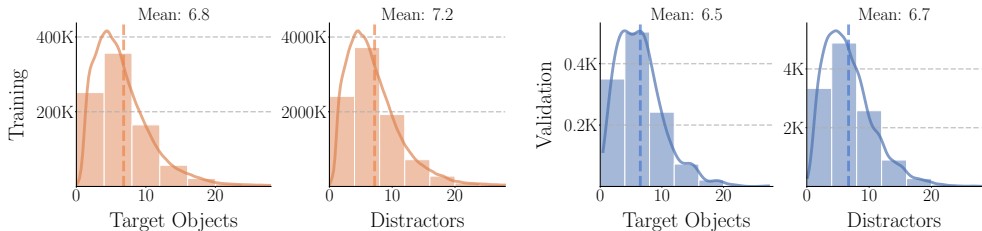

Figure E: Euclidean distances of the objects included in the episodes of training (orange) and validation (blue) splits of PInNED dataset. The plots consider the distances from the start position to the target object (left) and to all distractors (right). Distances are measured in meters, with the mean value for each plot displayed at the top.

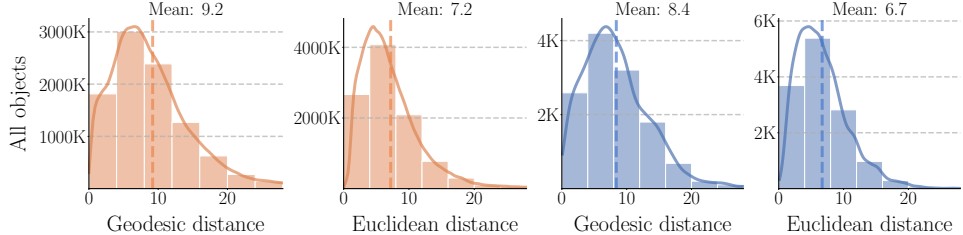

Figure F: Plots of the geodesic and Euclidean distances for all the objects placed in the episodes of PInNED dataset. Training (orange) and validation splits (blue) are presented in terms of distances from the start position to all the spawned additional objects. All the distances are plotted in meters, and the mean value of each plot is shown on top.

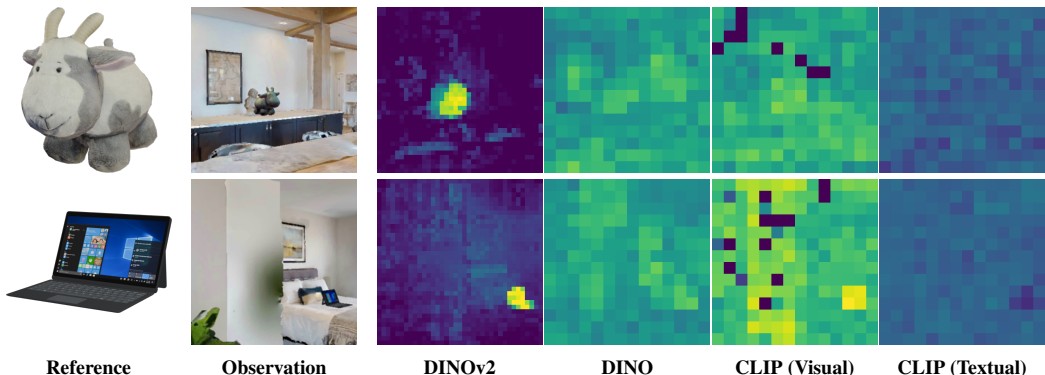

Figure G: Comparison of the similarities between the patch-level features on two observations of an agent extracted with different backbones, DINOv2, DINO, CLIP with visual references, and CLIP with textual references, and the references. Purple values represent low similarity values, while yellow values represent high similarity values.

have discrepancies in appearance. For methods based on semantic features, the similarity threshold is the key element in balancing confidence and caution.

In Table C, we report the average cosine similarities in the DINOv2 embedding space per category. In particular, we extracted the CLS token from each frontal goal image of the validation set and computed the cosine similarities against the other goal images from the same category (i.e. intra-category) and against goal images from different categories (i.e. inter-categories). The results show that the intra-category similarity presents a strong relation with the category error (CE) and category matches (CM) metrics. Indeed, the agent tends to mistake instances from categories with large intra-category similarity values, such as '*eyeglasses*', '*headphones*', and '*shoes*', while these mistakes are reduced in categories such as '*camera*' and '*toy*' that are characterized by a larger variability in their instances. When we adopt textual references as targets, the challenges concern how well multimodal spaces embed fine-grained details, and how similarity behaves accordingly. Previous work [7, 10] has shown that this challenge is non-trivial and still open. Our dataset represents a

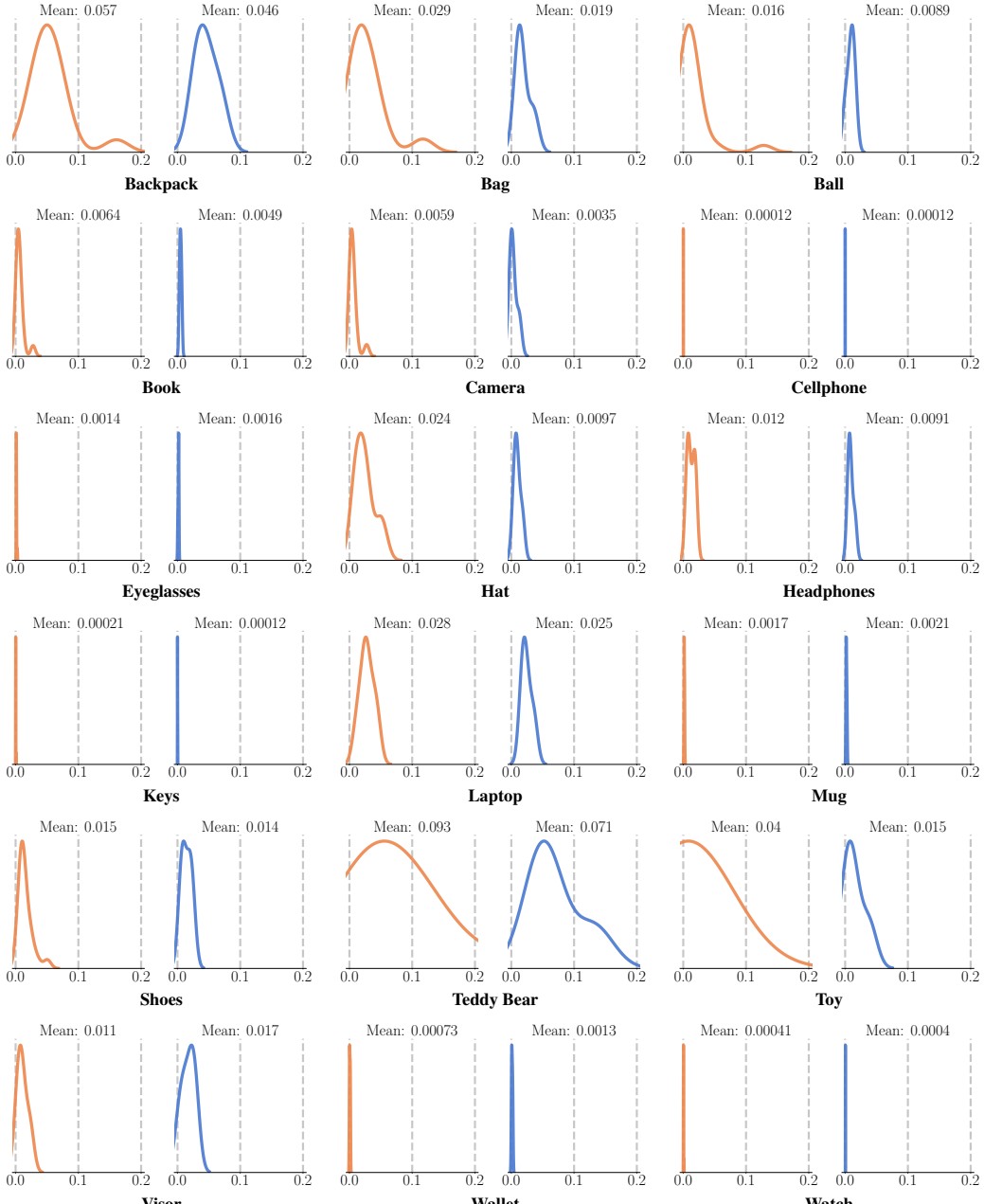

Figure H: Distribution of the volumes in meters of the bounding boxes of the objects in PInNED dataset. Two plots are shown for each semantic category, reflecting respectively the objects of training (orange) and validation (blue) splits. Each plot is accompanied by the corresponding mean of bounding box volumes of the objects in each split.

further step in this direction, providing a benchmark to evaluate the capabilities of visual-language models in recognizing fine-grained details. Future works can exploit our training set to instruct the models to distinguish instances of the same category by focusing on adjectives and attributes.

**Surfaces Details.** As described in Sec. 3.3, the spawning position of each object in the PInNED dataset is selected by sampling from the positions of a curated set of suitable surface macro-categories included in the semantic annotations of HM3D. The surface categories selected for the creation of the dataset are: *armchair, bed, bench, cabinet, piano, rug, sofa, table*. These surfaces are valid for all the object categories and there are no subsets of surfaces dedicated to specific categories. There are categories, especially '*shoes*', that are unlikely to be placed on certain surfaces. However, the scope

Table C: Navigation results of the modular agent that employs DINOv2 as the matching module on the validation episodes of PInNED dataset, considering the performance of the agent for each category. Moreover, we report the average intra-category and inter-category cosine similarities computed on the frontal goal images.

| | **Navigation Metrics** | | | | | **Detection Metrics** | | | | **Similarity** | |
| Category | SR↑ | SPL↑ | CE↓ | D2G↓ | Steps | %Match↑ | TM↑ | CM↓ | NM↓ | Intra-Category | Inter-Category |
|---|---|---|---|---|---|---|---|---|---|---|---|
| Backpack | 26.47 | 14.04 | 36.77 | 5.79 | 408.7 | 85.29 | 53.27 | 46.57 | 0.16 | 0.510 | 0.110 |
| Bag | 23.08 | 13.65 | 40.00 | 6.16 | 406.5 | 93.85 | 44.62 | 55.04 | 0.34 | 0.348 | 0.121 |
| Ball | 20.90 | 10.29 | 23.88 | 6.48 | 613.1 | 61.19 | 36.06 | 63.87 | 0.07 | 0.258 | 0.068 |
| Book | 19.40 | 10.83 | 35.82 | 5.71 | 484.3 | 86.57 | 58.51 | 40.16 | 1.33 | 0.613 | 0.106 |
| Camera | 7.46 | 3.38 | 7.50 | 8.57 | 883.2 | 20.90 | 69.23 | 23.08 | 7.69 | 0.152 | 0.050 |
| Cellphone | 8.96 | 3.11 | 14.92 | 8.63 | 844.8 | 32.84 | 7.81 | 90.96 | 1.23 | 0.506 | 0.112 |
| Eyeglasses | 10.45 | 5.08 | 32.83 | 7.70 | 682.0 | 62.69 | 79.80 | 19.95 | 0.25 | 0.846 | 0.104 |
| Hat | 26.87 | 11.95 | 23.88 | 6.45 | 652.8 | 67.16 | 88.08 | 11.89 | 0.03 | 0.549 | 0.084 |
| Headphones | 16.92 | 9.71 | 40.00 | 7.35 | 492.8 | 84.62 | 14.58 | 85.29 | 0.13 | 0.764 | 0.098 |
| Keys | 0.00 | 0.00 | 8.82 | 8.38 | 974.2 | 2.94 | 0.00 | 0.00 | 100.00 | 0.558 | 0.102 |
| Laptop | 21.54 | 11.50 | 49.23 | 7.01 | 455.3 | 93.85 | 16.86 | 82.60 | 0.54 | 0.348 | 0.084 |
| Mug | 10.61 | 4.47 | 10.61 | 8.10 | 911.8 | 22.73 | 92.00 | 4.50 | 3.50 | 0.298 | 0.073 |
| Shoes | 16.92 | 12.44 | 44.62 | 6.75 | 318.8 | 95.38 | 8.31 | 91.69 | 0.00 | 0.631 | 0.087 |
| Teddy Bear | 19.12 | 13.48 | 52.94 | 7.07 | 335.5 | 91.18 | 68.92 | 16.62 | 14.46 | 0.548 | 0.066 |
| Toy | 26.56 | 13.18 | 3.12 | 6.16 | 754.6 | 48.44 | 99.27 | 0.00 | 0.73 | 0.137 | 0.087 |
| Visor | 11.94 | 5.99 | 31.34 | 7.99 | 657.0 | 52.24 | 52.47 | 45.33 | 2.20 | 0.316 | 0.148 |
| Wallet | 0.00 | 0.00 | 6.15 | 8.39 | 985.3 | 1.54 | 0.00 | 0.00 | 100.00 | 0.282 | 0.105 |
| Watch | 0.00 | 0.00 | 7.69 | 8.39 | 999.0 | 0.00 | 0.00 | 0.00 | 100.00 | 0.566 | 0.102 |

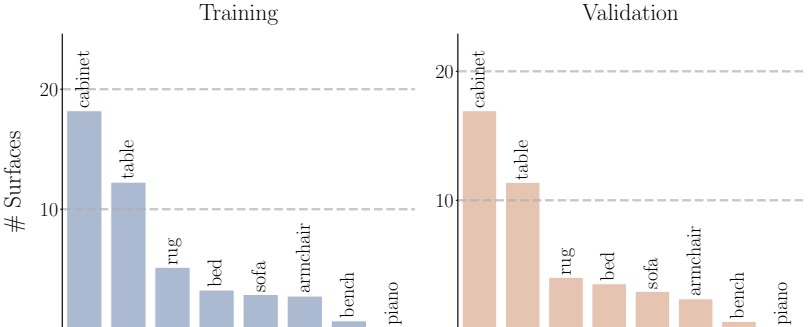

Figure I: Plot of the mean number of surfaces in each environment that are suitable for object placement in the training (left) and validation (right) splits of the PInNED dataset.

of the task is to have objects that could be placed everywhere and teach a robotic agent to find them. A teddy bear is not necessarily located on the bed, but could be located anywhere, even on the kitchen table. If we assume a real-world scenario in which a child forgets the teddy bear on the kitchen table, the agent should not go directly to the bedroom, but look for the object in the whole environment. This is the reason for which we adopted a consistent spawning mechanism across all the categories. We identify this combination of objects that could be placed everywhere and the consistent spawning mechanisms as the correct approach for providing a dataset covering a large set of possible real-world scenarios that avoid the exploitation of prior knowledge on the object placement.

In Fig. I, we showcase the occurrences of the suitable surfaces in the environments of HM3D [56]. Notably, the distribution of spawnable surfaces remains consistent between the training and validation splits. This implies a recurring pattern in the furnishing of indoor spaces contained in the HM3D dataset and used for the PIN task.

**Hard Detection Cases.** In Fig. J, we show four episodes in which detecting the target is particularly challenging. These targets belong, respectively, to the '*wallet*', '*camera*', '*watch*', and '*keys*' categories. Table C shows that these categories are the most challenging ones for the modular agent with DINOv2, which is the best-performing agent according to Table 2. Indeed, the categories '*keys*', '*wallet*', and '*watch*' all yielded no successful episodes. These objects are hard to detect even for a human, confirming how challenging the PIN task is. Future work should investigate the possibility of moving the agent closer to areas in which there are small objects that cannot be identified as the target from longer distances.

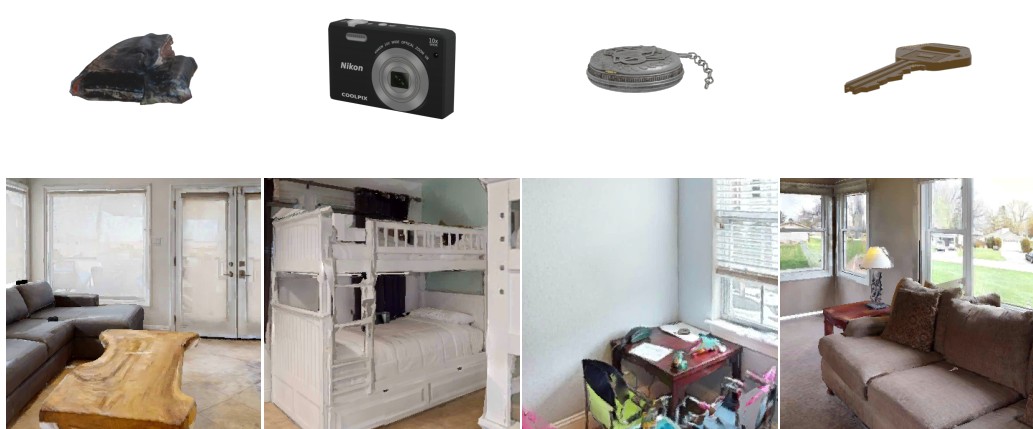

Figure J: Examples of situations in which detecting the target in the embodied environment is particularly challenging. We depict the frontal visual references of the target in the first row and a portion of an agent's observation containing the target in the second row.

Table D: Navigation results of the modular agent that employs SuperGlue as the matching module on the validation episodes of PInNED dataset, considering different resize values of the visual references of the target provided to the matching module.

| Resize | Navigation Metrics | | | | | Detection Metrics | | | |
|---|---|---|---|---|---|---|---|---|---|
| | SR↑ | SPL↑ | CE↓ | D2G↓ | Steps | %Match↑ | TM↑ | CM↓ | NM↓ |
| 360 | 2.51 | 0.82 | **7.05** | 8.48 | 881.6 | 17.77 | **43.76** | 5.17 | **51.07** |
| 180 | 3.02 | 1.20 | 7.21 | 8.48 | 864.1 | 20.70 | 21.72 | 3.58 | 76.35 |
| 180, 360 | **3.27** | **1.28** | 7.58 | **8.36** | **804.0** | **29.42** | 16.96 | **3.44** | 79.60 |

**Fine-Grained vs General Descriptions Comparison.** In Table E we present an ablation study in which we compare the performance of the baselines with both fine-grained and general object categories. Specifically, we conducted the following experiments. For the modular agents based on CLIP and OWL as the matching module, we leveraged the general object category (e.g. backpack) instead of the fine-grained textual descriptions as navigation targets, while maintaining the same similarity threshold. The results on both CLIP and OWL present similar behaviors: the number of successful episodes is slightly increased, but also the number of episodes in which the agent mistakes reaching distractors of the same category and the number of matches with them increased. Moreover, the reduction in the average number of steps indicates that similarities, on average, are higher. The increase in successful episodes is surprising but in line with the findings of previous works in the literature [7, 10], which demonstrate that current vision-language models struggle with fine-grained details. These results show that our work can help future works in the realization and evaluation of vision-language models with improved understanding capabilities of details. Concerning the end-to-end agent RIM, we trained the model on the CLIP embeddings extracted from the general object categories instead of the fine-grained textual descriptions. The results show a lower number of successful episodes and a higher number of episodes in which the agent reaches a distractor of the same category.

## F   Additional Implementation Details

**Modular Agents.** In Sec. 4.1, we introduce the modular agents tested on the PIN task. Their ability to distinguish a specific instance in a given observation depends on the score threshold that maximizes the detection results. We tune this threshold on a subset of the training episodes. For all the backbones except for SuperGlue [60], we extract two squared crops with size $360 \times 360$ from the $360 \times 640$ observation and resize them to the image resolutions on which the backbones have been trained. Then, we consider all the matches resulting from the two crops. At least a match over the threshold is required to consider the goal detected in an observation. For the textual modalities, we employ the

Table E: Navigation results on PInNED on the environments of HM3D dataset, comparing categorical and fine-grained textual modalities.

| | Backbone | Modality | Navigation Metrics | | | | | Detection Metrics | | | |
|---|---|---|---|---|---|---|---|---|---|---|---|
| | | | SR↑ | SPL↑ | CE↓ | D2G↓ | Steps | %Match↑ | TM↑ | CM↓ | NM↓ |
| *Modular Agents* | | | | | | | | | | | |
| CLIP [54] | ViT-B/16 | Categorical | 3.52 | 2.75 | 10.23 | 7.98 | 148.1 | 95.47 | 5.12 | 15.73 | 79.15 |
| CLIP [54] | ViT-B/16 | Fine-Grained | 3.10 | 1.82 | 9.31 | 7.94 | 503.1 | 62.95 | 20.07 | 22.07 | 57.86 |
| OWL [29, 49] | ViT-B/32 | Categorical | 7.79 | 3.73 | 19.96 | 7.96 | 780.6 | 38.81 | 26.50 | 58.49 | 15.01 |
| OWL [29, 49] | ViT-B/32 | Fine-Grained | 7.29 | 3.36 | 12.66 | 7.90 | 871.7 | 22.97 | 62.60 | 32.88 | 4.52 |
| *End-to-end Agents* | | | | | | | | | | | |
| RIM [18] | ResNet-50 | Categorical | 4.61 | 3.78 | 14.25 | 9.23 | 336.0 | - | - | - | - |
| RIM [18] | ResNet-50 | Fine-Grained | 7.12 | 6.67 | 10.44 | 8.43 | 409.3 | - | - | - | - |

80 prompt templates proposed by Radford *et al.* [54] for ImageNet [24]. In this section, we report additional implementation details for each backbone.

📌 **SuperGlue [60]**: We observe that SuperGlue struggles to match the visual references with the observations of the agent and that the resolution of the references influences the matching capabilities. In particular, we provide the visual references to SuperGlue as squared images $360 \times 360$, corresponding to the shortest side of the observation of the agent. For each visual reference, namely for each of the three views of the object, we provide two resizes of the object such that the longest side is, respectively, 360 and 180. This procedure results in two reference images for each view of the object, an image entirely occupied by the object and an image where the object occupies a quarter of it. In Table D we show that this approach results in a higher success rate than having a single image per object view. Moreover, we employ the *indoor* weights of SuperGlue with a threshold of 0.2 on the confidence of each matched keypoints pair and a matching threshold $\sigma$ of 8.0 on the confidence sum of all the matched keypoints pairs.

📌 **CLIP [54]**: We employ CLIP ViT-B/16 with the pre-trained weights from OpenAI for both the experiments with visual and textual references. We resize the two observation crops to $224 \times 224$, resulting in a grid of $14 \times 14$ patches. The best matching threshold $\sigma$ for the visual and textual modalities are, respectively, 0.575 and 0.28.

📌 **CLIP-Grad**: We follow the implementation of the network interpretability method proposed in CoW [29] on top of CLIP with textual references. We employ CLIP ViT-B/32 with the pre-trained weights from OpenAI and matching threshold 0.85.

📌 **OWL [49]**: OWL is an open-vocabulary detector that is trained in two steps: (i) a large contrastive image-text pre-training following LiT [78] and (ii) an object-level training on publicly available detection datasets (Open Images V4 [38], Objects 365 [63], and Visual Genome [37]). We employ a matching threshold of 0.25 applied to the predicted bounding box scores.

📌 **DINO [12]/DINOv2 [51]**: DINO is a self-supervised backbone pre-trained according to a self-distillation training paradigm. DINOv2 is an improved version of DINO with the aim of producing general-purpose visual features. We employ DINO ViT-B/16 and DINOv2 ViT-B/14 trained, respectively, on ImageNet-1k [24] and LVD-142M [51]. We use the same input image resolutions on which they are trained, namely $224 \times 224$ and $518 \times 518$, producing $14 \times 14$ and $37 \times 37$ grids of patches. The best matching scores are, respectively, 0.575 and 0.5.

📌 **PerSAM/PerSAM-F [80]**: We leverage the implementation of PerSAM on SAM ViT-B/16, trained on SA-1B, with input image resolution at $1,024$. PerSAM-F is a variant of PerSAM that fine-tunes the model on the reference image, We follow the training configuration of the original implementation. We consider the maximum patch-level similarity between the reference images and the observation crop as the matching score on which we apply the thresholds 0.925 and 0.61 for, respectively, PerSAM and PerSAM-F.

**End-to-End Agents.** As mentioned in Sec. 4.2 end-to-end approaches use a neural network policy which is trained end-to-end to directly process sensor observations and predict the atomic actions needed to fulfill the required task. In our case, we adapted two recent end-to-end approaches for ObjectNav finetuning them to perform PIN task: RIM [18] and ZSON [45].

📌 **RIM [18]**: The model is finetuned using behavior cloning following Chen *et al.* [18] approach and starting from the pre-trained weights for ObjectNav [6]. We evaluate two variants of the fine-tuned model, conditioned on visual features and conditioned on textual features. In RIM approach, besides the episodic implicit map that is updated recursively, the input of the policy at each timestep is composed of the concatenation of the features extracted from RGB and depth observation, the pose of the agent, previous action, and the target object category. To adapt RIM for the PIN task, we modify the features extracted from the object category label. Originally each label is associated with a row in a lookup table containing learnable embeddings of length 32. In our adaptation, we replace such embeddings with CLIP (ViT-B/16) features extracted using the visual or textual references. Since each input reference modality is described by 3 images or descriptions, we compute the mean of the features extracted from each reference. Following, a learnable linear layer is trained to project CLIP features to a vector of length 32. The resulting embedding is used to condition the navigation of the RIM agent. The fine-tuning process is performed on a single GPU for a total of $\approx$ 2M fine-tuning steps over $\approx$ 24 hours.

📌 **ZSON [45]**: For the adaptation of the ZSON method, we fine-tuned the model pre-trained on the ImageNav task, following the same approach as Majumdar *et al.* [45]. The agent is fine-tuned with reinforcement learning using an adaptation of ZSON reward but ignoring the angle to the goal since it is not a component considered in the PIN task. The resulting reward is $r_t = r_{success} - \Delta_{dtg} + r_{slack}$. We refer to Majumdar *et al.* [45] for a description of the components of the reward. Moreover, while the original approach uses ImageNav goals that are represented as photos captured at the position that the agent is required to reach, we used image references of the target instance to perform the fine-tuning. The model is fine-tuned on a single GPU for $\approx$ 24 hours for a total of $\approx$ 5M fine-tuning steps.

**Compute Information.** We performed our experiments on a computing platform composed of NVIDIA RTX5000 GPUs and 8 GB of CPU memory for each job. A job can be computed on a single GPU. Each episode step for the modular agents requires an average of $\approx$ 200ms to be executed. Hence, the entire DINOv2 experiment on the $1,193$ validation episodes, with an average number of steps equal to $658.7$, requires $\approx$ 44 computation hours. The entire evaluation on the validation split for the end-to-end agents requires $\approx$ 5 computation hours.

## G   Licenses and Terms of Use

The episodes of the PInNED dataset are built using the scenes from the HM3D dataset [56]. The scenes of the HM3D dataset are released under the Matterport End User License Agreement, which permits non-commercial academic use.

For the augmentation of HM3D scenes with additional objects, PInNED dataset utilizes 3D object assets from Objaverse-XL dataset [22]. Objaverse-XL is distributed under the ODC-By 1.0 license, with individual objects retrieved from various sources, including GitHub, Thingiverse, Sketchfab, Polycam, and the Smithsonian Institution. Each object is subject to the licensing terms of its respective source, necessitating users to evaluate license compliance based on their specific downstream applications.

Nevertheless, the specific objects included in our dataset are restricted to assets sourced from Sketchfab which are released under various Creative Commons licenses. Specifically, the dataset includes assets under the following licenses: CC BY (311 objects), CC BY-NC (14 objects), CC BY-SA (8 objects), CC BY-NC-SA (3 objects), and CC0 (2 objects).

The episodes of the PInNED dataset, along with the manually annotated object descriptions are released under the CC BY license, while the codebase for the PIN task is released under the MIT license.

The authors accept full responsibility for any rights violations arising from the use or publication of the data and content in this paper. All licenses related to external content included in this paper ensure no infringement on third-party rights.

```
1  {
2      "episode_id": "0",
3      "scene_id": "hm3d/val/00800-TEEsavR23oF/TEEsavR23oF.basis.glb
           ",
4      "start_position": [-0.28, 0.013, -6.54],
5      "start_rotation": [0, 0.98, 0, 0.20],
6      "info": {"geodesic_distance": 8.24},
7      "goals": [
8          {
9              "object_category": "backpack",
10             "object_id": "3f5948f7f47343acb868072a7fe92ada",
11             "position": [-5.13, 1.08, -0.81]
12         }
13     ],
14     "distractors": [
15         {
16             "object_category": "backpack",
17             "object_id": "3c47af8b6a3e413f94c74f86d4c396ed",
18             "position": [-3.46, 2.20, -4.30]
19         },
20         {
21             "object_category": "backpack",
22             "object_id": "0b795895343b44b69191ef9b55b35840",
23             "position": [-11.17, 0.88, -0.36]
24         },
25         {
26             "object_category": "backpack",
27             "object_id": "d86ee61984544b45a9f11f49e5e02c43",
28             "position": [-9.13, 1.22, -3.52]
29         },
30         {
31             "object_category": "mug",
32             "object_id": "d26e9bfce2644bb7af6710c6511ea718",
33             "position": [-7.84, 0.62, -0.14],
34         },
35         {
36             "object_category": "laptop",
37             "object_id": "6495988c6c044c76a2fc9f9278543c16",
38             "position": [-1.64, 0.87, -6.15],
39         },
40         {
41             "object_category": "headphones",
42             "object_id": "ccf60b0502784fb38e483a6b07cfad53",
43             "position": [3.41, 0.84, -8.21],
44         },
45     ],
46     "scene_dataset_config": "data/scene_datasets/hm3d/hm3
           d_annotated_basis.scene_dataset_config.json",
47     "object_category": "backpack",
48     "object_id": "3f5948f7f47343acb868072a7fe92ada"
49 }
```

Listing A: Python dictionary containing a sample of the episodes contained in PInNED dataset. The list of distractors is skimmed for better visualization.

# H   Assets

The episodes of PInNED dataset are defined as Python dictionaries containing relevant information for the execution of the PIN task with the Habitat simulator. An example of episode annotation is presented in Listing A. Each episode specifies the environment where it is taking place, the starting position and rotation of the agent, along with the position and object identifier of the target instance and the distractors.

```
1  {
2      "scale": [0.116, 0.116, 0.116],
3      "render_asset": "0a96f1f19afc432bb22c3d74da546338.glb",
4      "requires_lighting": true,
5      "up": [0.0, 1.0, 0.0],
6      "front": [0.0, 1.0, 0.0],
7      "COM": [0.0, 0.0, 0.0],
8      "gravity": [0, 0, 0],
9      "force_flat_shading": true,
10     "is_collidable": true,
11     "use_mesh_for_collision": true,
12     "semantic_id": 2,
13     "semantic_category": "ball"
14 }
```

Listing B: Python dictionary containing the information used by Habitat simulator to instantiate a specific object instance in the environment.

The information used by the Habitat simulator to resize and instantiate each 3D object at the position specified by the episodes of PInNED dataset is also contained in a Python dictionary, where a specific file represents each object. A sample of object annotation is showcased in Listing B.

# I Datasheet

In this section, we present a comprehensive datasheet [30] for the proposed dataset, providing a unified reference for relevant information on the PInNED episodes and the objects used to build the dataset.

## I.a Motivation

**For what purpose was the dataset created?** The PInNED dataset has been built with the motivation of fostering future research on smart navigation agents. Such agents need to acquire the capability of distinguishing between different instances of the same object category and leverage different modalities of inputs to reach a specific object asked by the user. The dataset introduces a novel task in Embodied AI research and, in order to run the episode of the PInNED dataset, the Habitat simulator needs to be used. Instructions on how to run and instantiate the episodes of PInNED dataset are included in the public repository described in Sec. A.

**Who created the dataset and on behalf of which entity?** The dataset was created by researchers at the University of Modena and Reggio Emilia.

**Who funded the creation of the dataset?** Refer to the Acknowledgments and Disclosure of Funding section in the main paper.

## I.b Composition

**What do the instances that comprise the dataset represent?** The PInNED dataset consists of generated navigation episodes designed to address the PIN task, accompanied by a list of object identifiers used in each episode within the Habitat simulator. As the dataset is composed of navigation episodes, containing all necessary information for the simulator to execute the task, no additional metadata is provided. However, an example of episode annotations is included in Listing A.

**How many instances are there in total?** The dataset of episodes for the PIN task is composed of a total of $865,519$ training episodes and $1,193$ validation episodes. Moving on to the objects contained in the PInNED dataset, the total number of unique object instances that are injected in the navigation environments is $338$.

**Does the dataset contain all possible instances or is it a sample (not necessarily random) of instances from a larger set?** While episodes of the PInNED dataset are generated procedurally by the authors of the paper, the objects used as additional objects are part of the objects released from

Objaverse-XL dataset [22], which is composed of 3D models from different online sources such as GitHub, Thingiverse, Sketchfab, Polycam, and the Smithsonian Institution. The objects of PInNED are however restricted to 3D models included in Sketchfab.

**What data does each instance consist of?** The dataset content is defined by the information of the episodes for the PIN task. Each episode is represented as a dictionary containing the information needed by the Habitat simulator [61] to execute the task. A *.json* file including a list of the navigation episodes is produced for each scene included in HM3D dataset. We refer to Listing A for a sample of episode annotation. Each episode in the dataset specifies additional objects that are placed at a specific location loading *.glb* files containing the meshes of the objects. The *.glb* files used to instantiate the episodes of the PInNED dataset are downloadable from Objaverse-XL API using the Python script provided in the codebase. Each 3D object is associated with a *.json* file containing a dictionary with the information needed by the Habitat simulator to correctly instantiate the object in the environment in terms of size and appearance.

**Is there a label or target associated with each instance?** Each object used for the PInNED dataset is manually associated with an object category label to correctly perform the placement procedure of distractors belonging to the same category of the target instance, as well as computing metrics related to the PIN task. However, the object category label should not be used by the agent to tackle the PIN task. For each episode, only one instance is defined as the correct target to complete the task successfully.

**Are there recommended data splits?** The episodes of the PInNED dataset are divided into training and validation splits depending on the environment where the episodes are taking place. The environments are divided into training and validation splits following the environmental-level division performed by Ramakrishnan *et al.* [56]. Regarding the additional objects included in PInNED dataset, the object instances are divided into 266 training instances and 72 validation instances. It is worth noting that the sets of instances used for the training and validation splits do not overlap.

**Are there any errors, sources of noise, or redundancies in the dataset?** The additional objects on the surfaces of HM3D environments could be misplaced due to noise in the original annotations of the scene, or due to the presence of clutter at the acquisition time of the environment. Other sources of noise could be related to possible typos in the process of annotation of the textual descriptions of the additional objects.

**Is the dataset self-contained, or does it link to or otherwise rely on external resources?** The PInNED dataset relies on the scenes included in the HM3D dataset of 3D spaces and on the 3D object assets included in the Objaverse-XL dataset.

**Does the dataset contain data that might be considered confidential? Does the dataset contain data that, if viewed directly, might be offensive, insulting, threatening, or might otherwise cause anxiety?** No confidential or disturbing data is contained in the content of PInNED dataset.

## I.c    Collection Process

**How was the data associated with each instance acquired? What mechanisms or procedures were used to collect the data?** The additional objects used for the PInNED dataset are manually selected using the Python API from Objaverse-XL dataset.

The generation of the visual references of the target objects has been performed using Blender, where the 3D mesh of the object is rendered and captured in an isolated setting. The camera performs a 30-degree yaw rotation around the object to capture a favorable view of the objects. Then, each instance is rotated by 180 degrees in yaw to view its reverse side, while a 90-degree pitch rotation is used to observe the upper side of the object. This procedure produces three visual references for each target object.

The process of annotation of the textual descriptions of each object is performed by the authors of the paper. Two objects of the same object category are shown to each annotator that is required to describe one of the two instances in a way that is distinguishable from the other. The final procedure used three annotators, for a total of three textual descriptions for each object. Samples of the input references related to the objects of PInNED dataset are shown in Fig. C and Fig. D.

The episodes of PInNED are generated by spawning the selected additional objects in the scene after extracting all suitable surfaces from the semantic annotation of the environment. We refer to Sec. 3.3 for more details on object placement.

**If the dataset is a sample from a larger set, what was the sampling strategy?** The sampling strategy for selecting the objects from Objaverse-XL is based on a manual assessment of the photo-realistic properties of the selected objects and the corresponding visual appearance of the object when rendered using the Habitat simulator. The sampling strategy of the objects contained in the episodes of PInNED dataset is a random sampling. For each episode, a goal object category is selected, and a specific target instance is sampled from the set of suitable objects. Instances belonging to the same object category of the target object are sampled and positioned in the environment as distractors. If other spawnable surfaces are available, more distractors belonging to other object categories are placed in the environment. Details on the number of additional objects placed in the episodes of PInNED dataset are included in Sec. D. For the final generation of the episodes of PInNED dataset, 400 episodes are generated for each possible object category on the environments of the training split, while 2 episodes for each object category are generated in every environment of the validation split.

**Who was involved in the data collection process?** The actors performing the data collection and annotation process of the dataset are the authors of the paper.

**Over what timeframe was the data collected?** The dataset assets were collected and the episodes of the PInNED dataset were generated between November 2023 and May 2024.

**Were any ethical review processes conducted?** No ethical review process was necessary for the collection of the dataset.

### I.d   Preprocessing / Cleaning / Labeling

**Was any preprocessing/cleaning/labeling of the data done?** The objects used in the PInNED dataset are manually resized when rendered with the Habitat simulator adjusting their dimension compared with the surrounding environment to be similar to their real-world counterpart. Each 3D object is associated with a corresponding object category label to allow the usage of different instances of the same object category when tackling the PIN task. The episodes of PInNED dataset are, instead, validated using the Habitat simulator to remove any episode containing objects that are not reachable from the starting position of the agent.

### I.e   Uses

**Has the dataset been used for any tasks already?** The PInNED dataset can be used to train and evaluate agents for the Personalized Instance-based Navigation (PIN) task. See Sec. 3 and Sec. 5 for more details on the task definition and the experimental evaluation.

**What (other) tasks could the dataset be used for?** The dataset could be used for other tasks involving recognition or manipulation on specific instances using visual or textual references as input.

**Is there anything about the composition of the dataset or the way it was collected and pre-processed/cleaned/labeled that might impact future uses?** Users need to follow and respect the licenses associated with the additional 3D objects and the episodes contained in this dataset.

### I.f   Distribution

**How will the dataset will be distributed?** The dataset is made public through the release of a public GitHub repository. The repository containing dataset and codebase is released at this url: https://github.com/aimagelab/pin.

**When will the dataset be distributed?** The dataset has been publicly released on October 2024.

**Will the dataset be distributed under a copyright or other intellectual property (IP) license, and/or under applicable terms of use (ToU)?** The dataset and the object annotations are released under the CC BY license. The codebase is released under the MIT license. The additional objects contained in the episodes of PInNED dataset are subject to the licenses that they are released under.

**Have any third parties imposed IP-based or other restrictions on the data associated with the instances?** Any restrictions are related to additional objects and to the licenses which they are released under. Users need to assess license questions based on their use.

### I.g Maintenance

**Who will be supporting/hosting/maintaining the dataset?** The dataset will be maintained by the authors of the paper who commit to maintaining the dataset long-term.

**How can the owner/curator/manager of the dataset be contacted?** The authors can be contacted at `{firstname.lastname}@unimore.it`.

**Will the dataset be updated?** A potential future update could involve extending the dataset to include a test split, upon receiving permission from the HM3D dataset owners to access the environments in the test split.

**If others want to extend/augment/build on/contribute to the dataset, is there a mechanism for them to do so?** Users are free to extend the dataset at the condition of following and respecting the licenses associated with the dataset and associated additional objects by contacting the authors on the public repository.