# OpenReview forum: "Personalized Instance-based Navigation Toward User-Specific Objects in Realistic Environments"
_NeurIPS.cc/2024/Datasets_and_Benchmarks_Track — NeurIPS 2024 Track Datasets and Benchmarks Poster_

### Official Review · Reviewer_6hP2 · 2024-07-23
**Personalized Instance-based Navigation dataset**

**Rating:** 7
**Confidence:** 2
**Correctness:** I think the claims are appropriate.
**Clarity:** Good.

**Review:**

The experimental section presents evaluation metrics and results. The navigation results demonstrate its effectiveness.

**Strengths:**

The paper proposes and advances Personalized Instance-based Navigation, presenting adequate novelty.

**Additional Feedback:**

I have no further feedback.

**Documentation:**

The paper provides sufficient details on the data.

**Ethics:**

I have no apparent ethical concerns regarding this paper.

**Limitations:**

I have no more concerns about limitations.

**Opportunities For Improvement:**

The dataset includes 338 object instances. I am wondering whether the similarity of objects can influence the performance, or whether object similarity can be used to improve the performance.

**Relation To Prior Work:**

The differences from previous works are clearly clarified.

**Summary And Contributions:**

The paper proposes a novel task of Personalized Instance-based Navigation, which aims to locate and reach a specific personalized target. Additionally, a personalized instance-based navigation embodied dataset, containing 338 object instances, is built and released.

---

> ### Author Rebuttal · Authors · 2024-08-16
>
> We thank the Reviewer for dedicating time to review our paper, appreciating the novelty of the proposed task and accompanying dataset, and acknowledging the quality of the experimental evaluation. Below, we address the question raised in the review.
>
> **The authors should discuss whether the similarity of objects can influence the performance, or whether object similarity can be used to improve the performance.**
>
> We thank the Reviewer for raising this interesting question. The similarity of objects is indeed a critical factor in the PIN task. The presence of distractors increases the challenge of the proposed task, as the agent must balance between being overly cautious and overly confident when identifying target instances. This trade-off is central to the effectiveness of the navigation approaches.
>
> In particular, concerning images as references of the target object, re-identification methods should be a robust solution against distractors due to considering the matching between keypoints instead of the semantic similarity between observation and reference. Indeed, in Table 2 of the main paper, the state-of-the-art re-identification method SuperGlue has a lower category error than DINOv2 and CLIP. However, it presents the worst results according to SR and SPL, showing difficulties in matching keypoints when observation and reference have discrepancies in appearance. For methods based on semantic features, the similarity threshold is the key element in balancing confidence and caution.
>
> In the attached PDF, we report a revised version of Table C from the Appendix, in which we added the average cosine similarities in the DINOv2 embedding space per category. In particular, we extracted the CLS token from each frontal goal image of the validation set and computed the cosine similarities against the other goal images from the same category (*i.e.* intra-category) and against goal images from different categories (*i.e.* inter-categories). The results show that the intra-category similarity presents a strong relation with the category error (CE) and category matches (CM) metrics. Indeed, the agent tends to mistake instances from categories with large intra-category similarity values, such as *eyeglasses*, *headphones*, and *shoes*, while these mistakes are reduced in categories such as *camera* and *toy* that are characterized by a larger variability in their instances.
>
> When we adopt textual references as targets, the challenges concern how well multimodal spaces embed fine-grained details, and how similarity behaves accordingly. Previous work [A, B] has shown that this challenge is non-trivial and still open. Our dataset represents a further step in this direction, providing a benchmark to evaluate the capabilities of visual-language models in recognizing fine-grained details. Future works can exploit our training set to instruct the models to distinguish instances of the same category by focusing on adjectives and attributes.
>
> [A] Bianchi L., Carrara F., Messina N., Gennaro C., & Falchi F., "*The devil is in the fine-grained details: Evaluating open-vocabulary object detectors for fine-grained understanding*" in CVPR 2024.
>
> [B] Bravo M. A.,  Mittal S., Ging S., & Brox T., "*Open-vocabulary Attribute Detection*", in CVPR 2023.

---

### Official Review · Reviewer_kDUs · 2024-07-23
**Review of Submission 1647**

**Rating:** 6
**Confidence:** 3
**Correctness:** Yes.
**Clarity:** Paper is well written and easily foll…

**Review:**

- Quality:

  - Pros:
    - The work introduces a novel task, PIN, addressing personalized object navigation in realistic environments.
    - The creation of the PInNED dataset provides a valuable resource for training and evaluating agents on the PIN task.
    - The evaluation of existing navigation approaches on the PIN task showcases the challenges and limitations of current methods.
  - Cons:
    - The paper could provide more detailed information on the experimental setup and methodology.
    - The limitations of the dataset and approaches used could be further discussed to provide a more comprehensive analysis.

- Clarity:

  - Pros:
    - The paper effectively introduces the PIN task and the PInNED dataset, providing clear definitions and explanations.
    - The use of figures and examples enhances the understanding of the task and dataset.
  - Cons:
    - Some sections could benefit from more detailed explanations, especially regarding the experimental results and analysis.

- Originality:

  - Pros:
    - The work introduces a unique task of personalized object navigation, which differs from traditional object-driven navigation tasks.
    - The creation of the PInNED dataset with personalized objects adds originality to the research domain.
  - Cons:
    - While the task and dataset are original, more discussion on the novelty of the approaches used could enhance the paper's originality.

- Significance:

  - Pros:
    - The work addresses an important gap in embodied navigation research by focusing on personalized object search.
    - The evaluation of existing methods on the PIN task highlights the need for further research in this area.
  - Cons:
    - The significance of the work could be further emphasized by discussing potential real-world applications and implications of the findings.

- Overall Assessment:

  - The work on Personalized Instance-based Navigation (PIN) and the PInNED dataset makes a valuable contribution to the field of embodied navigation. While the paper introduces a novel task and dataset, there is room for improvement in providing more detailed experimental information, discussing limitations, and emphasizing the significance of the work in real-world applications. Further research and development in this area could lead to advancements in personalized object search and navigation tasks.

**Strengths:**

1. **Significance of the Contribution**: The introduction of the PIN task is a significant advancement in the field of embodied AI. It addresses the gap in existing navigation tasks that typically focus on general object categories rather than personalized instances. This shift towards personal object navigation is crucial for developing more sophisticated and user-centric AI systems that can operate in real-world environments.

2. **Relevance to the Broader Research Community**: The proposed PInNED dataset and the PIN task are highly relevant to researchers working in areas such as robotics, computer vision, and human-robot interaction. By providing a benchmark for personalized navigation, the work encourages further exploration and innovation in embodied AI, potentially leading to improved applications in assistive technologies, smart homes, and autonomous systems.

3. **Quality of the Research**: The research demonstrates a solid methodological framework, including the design of the dataset and the evaluation of existing navigation agents. The paper outlines clear experimental setups and metrics for assessing performance, which enhances the reproducibility and reliability of the findings. The comparative analysis of modular and end-to-end approaches provides valuable insights into the strengths and weaknesses of current methodologies.

4. **Ethical and Social Implications**: The focus on personalized navigation has important ethical and social implications. By enabling AI systems to recognize and navigate towards personal objects, the research can enhance user experience and accessibility, particularly for individuals with disabilities or those requiring assistance in their environments. Additionally, the work raises considerations about privacy and data security, as personalized navigation systems may involve sensitive user data.

Overall, the submission's strengths lie in its innovative approach to a relevant problem, its methodological rigor, and its potential positive impact on society, making it a valuable contribution to the field of embodied AI.

**Additional Feedback:**

See comments above.

**Documentation:**

Welcome to offer more detailed documentations.

**Ethics:**

I do not suspect.

**Limitations:**

The authors have acknowledged some limitations of their work, but there are areas where they could improve their discussion of these limitations and potential negative societal impacts. Here are some constructive suggestions for improvement:

1. **Comprehensive Limitations Section**: The authors should include a dedicated section that explicitly outlines the limitations of their study. This section should address not only the scope of the PIN task and the dataset but also the assumptions made during the research. For example, they could discuss how the dataset's composition might affect the generalizability of the results and the potential biases introduced by the selection of personal objects.

2. **Discussion of Dataset Diversity**: The authors should elaborate on the diversity of the PInNED dataset. They could provide insights into how the dataset was constructed, including the variety of personal objects and environments represented. Acknowledging any gaps in this diversity and discussing how it might impact the performance of navigation agents in real-world scenarios would strengthen their analysis.

3. **Ethical Considerations**: The authors should expand their discussion on ethical implications, particularly regarding privacy and data security. They could outline the measures taken to protect user data and ensure that the use of personal objects does not lead to privacy violations. Additionally, they should consider discussing the potential for bias in the dataset and how it might affect different user groups, emphasizing the importance of fairness in AI systems.

4. **Potential Negative Societal Impacts**: The authors should explicitly address potential negative societal impacts of their work. For instance, they could discuss how the technology could be misused or lead to unintended consequences, such as surveillance or discrimination. Providing a balanced view of both the positive and negative implications of their research would demonstrate a thorough understanding of the societal context in which their work operates.

5. **Suggestions for Future Research**: The authors could conclude their limitations section with suggestions for future research that could address these limitations. For example, they might propose developing more diverse datasets, exploring user-agent interactions, or investigating the implications of their work in real-world applications.

By addressing these points, the authors would not only enhance the transparency and rigor of their research but also contribute to a more responsible discourse around the ethical and societal implications of their work. Acknowledging limitations and potential impacts is a sign of scholarly integrity and can foster trust within the research community and the public.

**Opportunities For Improvement:**

The submission on Personalized Instance-based Navigation (PIN) also has several limitations that can be examined along the same axes:

1. While the introduction of the PIN task is significant, it may still be limited in scope. The task primarily focuses on navigating to specific instances of objects, which may not fully capture the complexities of real-world navigation scenarios where multiple factors, such as dynamic environments and varying user preferences, come into play. This limitation could restrict the applicability of the findings to broader navigation challenges.

2. Although the PIN task is relevant, it may not address all aspects of personalized navigation that researchers in the field are interested in. For instance, the task does not consider the interaction between the agent and the user or the contextual understanding of the environment, which are critical for developing more advanced and user-friendly AI systems. This could limit its impact on the broader research community that seeks to integrate more holistic approaches to navigation.

3. While the research methodology is solid, there may be limitations in the dataset itself. The PInNED dataset, although extensive, may not encompass the full diversity of personal objects and environments that users encounter in real life. This could lead to a lack of generalizability of the results, as the performance of navigation agents may vary significantly in different contexts or with different object types that were not included in the dataset.

4. The focus on personalized navigation raises ethical concerns, particularly regarding privacy and data security. The use of personal objects implies that sensitive user data may be involved, and the paper may not sufficiently address how this data is handled, stored, and protected. Additionally, there may be implications related to bias in the dataset, as the selection of personal objects could reflect specific demographics or preferences, potentially leading to unequal performance across different user groups.

In summary, while the work presents valuable contributions to the field, its limitations include a potentially narrow focus on specific navigation tasks, a dataset that may not fully represent real-world diversity, and ethical considerations regarding privacy and bias that need to be more thoroughly addressed.

**Relation To Prior Work:**

Yes.

**Summary And Contributions:**

This paper introduces the novel task of Personalized Instance-based Navigation (PIN) where an agent navigates through environments to find a specific personal object. The key contributions of the submission are:

**New Task**: Introducing the PIN task where the agent must locate a specific personal object in a realistic environment.

**Dataset Creation**: Developing the Personalized Instance-based Navigation Embodied Dataset (PInNED) containing photo-realistic scenes with personalized objects.

**Benchmarking and Evaluation**: Benchmarling and assessing object-driven approaches of both modular agnets and end-to-end agents on the PIN task using the PInNED dataset.

These contributions aim to advance research in embodied navigation tasks by focusing on personalized object search in complex environments.

---

> ### Author Rebuttal · Authors · 2024-08-16
>
> We thank the Reviewer for the time devoted to our paper, providing insightful feedback, and acknowledging the relevance of the newly proposed task and dataset, as well as the experimental analysis included in the paper. In the following, we address the concerns pointed out in the review.
>
> **The paper could provide more detailed information on the experimental setup and methodology.**
>
> We thank the Reviewer for the suggestion. In addition to Sec. 4 and 5, further details about baselines and the experimental setup are in Sections D, E, and F of the Supplementary. If the Reviewer would like any specific aspect to be elaborated in more detail, we are willing to include them.
>
> **While the task and dataset are original, more discussion on the novelty of the approaches is needed. Also, potential real-world applications should be discussed.**
>
> We would like to clarify that the primary contributions of our paper lie in the introduction of the novel task and dataset. The approaches used in our experiments are existing methods from the literature in embodied navigation, which we adapted for the evaluation on the proposed task. Regarding real-world applications, a discussion is included in Sec. C of the Supplementary: we believe that the proposed benchmark can lead to more capable robotic assistants and autonomous systems, especially in household settings. Moreover, our dataset will serve as a benchmark for the development of novel algorithms in object-driven navigation.
>
> **While the introduction of the PIN task is significant, it might not fully capture the complexities of real-world navigation scenarios where dynamic environments and varying user preferences come into play. Also, it may not address all aspects that researchers are interested in, such as the interaction between the agent and the user or the contextual understanding of the environment.**
>
> We thank the Reviewer for this insightful feedback. Our dataset takes a step in the mentioned direction by considering dynamic environments, as the additional objects injected into the scenes are moved between different navigation episodes. Additionally, user preferences are implicitly addressed through the choice of input references provided at the beginning of each episode, guiding the agent toward specific instances. Finally, the target instances are provided to the agent as a series of images or as textual descriptions, covering both the real-world scenarios in which the user has photos of the target object or describes it through language. Regarding agent-user interaction, this is currently beyond the scope of our work. Additionally, we intentionally designed the task to have the agent rely on input references rather than contextual cues to identify specific instances that may be moved to multiple locations.
>
> **The PInNED dataset, although extensive, may not encompass the full diversity of personal objects and environments. This could lead to a lack of generalizability of the results.**
>
> We thank the Reviewer for raising this concern. We aimed to select objects to maximize intra-category diversity. Additionally, the task requires aligning references with observed objects in the scene, which is designed to enhance generalizability. For instance, our setting is intended to be more generalizable than traditional object-oriented navigation (ObjectNav), which involves finding any instance of fixed object categories. By focusing on the ability to locate target instances, we aim for approaches that are robust and generalizable, extending to different contexts and object types beyond those explicitly included in the dataset.
>
> **The focus on personalized navigation raises ethical concerns, regarding privacy and data security. Sensitive user data may be involved, and the paper may not sufficiently address how data is protected. Additionally, there may be implications related to bias, as the selection of personal objects could reflect specific demographics.**
>
> We thank the Reviewer for highlighting these important ethical concerns. Our dataset does not involve sensitive or personal data. As detailed in Sec. G of the Supplementary, all objects and environments featured in the dataset are used with permission from their owners and appropriate licenses. Furthermore, we conducted manual checks on the selected objects to ensure the absence of ethical issues or privacy concerns. We will further clarify this aspect in the Supplementary. Also, we will expand the discussion on the limitations to address potential biases. Indeed, the current dataset predominantly features objects typical of Western cultures, which may not capture variations from other cultures. We will consider extending the dataset with an out-of-distribution split with objects from different cultures to evaluate whether the approaches for personalized navigation generalize to a broader range of scenarios.
>
> **The authors should explicitly address the potential negative societal impacts of their work. For instance, they could discuss how the technology could be misused or lead to unintended consequences, such as surveillance or discrimination. Also, they could conclude their limitations section with suggestions for future research that could address these limitations.**
>
> We thank the Reviewer for highlighting these important aspects. We acknowledge that the dataset itself could be misused if users do not adhere to the licenses under which it has been released. However, concerns such as surveillance and discrimination are not applicable in this context, as the dataset does not involve personal data. We will include this discussion to provide a more comprehensive view of the potential impacts. Also, we will include suggestions for future research in the limitations section: these will cover future work for an additional split with out-of-distribution objects to evaluate the generalizability of the approaches, and allowing user-agent interactions during navigation.

---

### Official Review · Reviewer_UcJy · 2024-07-24

**Rating:** 6
**Confidence:** 5
**Correctness:** The claims made in the submission are…
**Clarity:** The paper is well written

**Review:**

**Quality:** The quality of the paper is satisfactory, meeting the standard scientific criteria.

**Clarity:** The paper clearly describes its methodologies and findings.

**Originality:** The work is a derivative extension based on existing datasets.

**Significance:** It addresses essential challenges that are inevitable in the progression of navigation problems.

**Strengths:**

The authors enhance realism and complexity of the existing dataset by introducing 338 additional objects, each accompanied by both visual and descriptive references, enriching the potential for diverse navigation challenges. Furthermore, the authors provide a thorough comparative analysis of mainstream navigation methods, effectively demonstrating the heightened difficulty and unique challenges presented by their dataset. This comparison not only highlights the dataset's robustness but also underscores the necessity for advanced methods in the field of personalized object navigation.

**Additional Feedback:**

No

**Documentation:**

The authors provide sufficient detail on data sources.

**Limitations:**

I believe that the authors have fully discussed the limitations.

**Opportunities For Improvement:**

1. **Dataset Comparison**: The comparison of datasets, particularly in Table 1 regarding the capabilities of AI2-THOR and RoboTHOR in providing visual references and supporting instance goal navigation, might have different understandings. It would be beneficial if the authors could provide a detailed explanation of the content and columns in the comparison table within the article to resolve any ambiguities.

2. **Object Placement Logic**: The authors do not detail the distribution logic for the added objects within the environments, which might affect the realism and applicability of the dataset for practical navigation scenarios. For example, the placement of objects like backpacks on tables or beds might be logical, whereas shoes appearing on beds might not.

3. **Scalability and Random Placement**: There is no discussion on the scalability of the dataset with the random placement of objects. If objects are placed randomly without logical constraints, it could lead to unrealistic scenarios and potentially infinite scalability, which might not effectively mimic real-world conditions.

4. **Comparison with ProcTHOR**: I believe the analysis could be enriched by comparing it with the ProcTHOR dataset, which comprises 1,633 interactive household objects across 108 categories. Could the authors consider adding this comparison to highlight potential strengths or gaps, especially regarding object interaction capabilities and category coverage?

**Relation To Prior Work:**

The paper discussed the differences from previous work.

**Summary And Contributions:**

The article presents a task termed Personalized Instance-based Navigation (PIN), where an autonomous agent is tasked with identifying and navigating towards a specific object in photorealistic environments, distinguishing it from similar items. The paper introduces the PInNED dataset, designed specifically for this task, which features a mix of photorealistic scenes augmented with additional 3D objects to simulate a more realistic and challenging navigation scenario.

---

> ### Author Rebuttal · Authors · 2024-08-16
>
> We would like to thank the Reviewer for the time devoted to our paper, providing insightful feedback and for appreciating our contributions. In the following, we address the concerns raised in the review.
>
> **The authors could provide a detailed explanation of the content and columns in the comparison of datasets (Table 1).**
>
> We thank the Reviewer for this suggestion. Table 1 presents a comparison of the main datasets available in the literature. Following, we describe the reported columns:
> - *Photo-Realistic Scenes*: the presence of photo-realistic scans taken from real-world environments (e.g. the scenes of HM3D are built from scans of real environments, while scenes in AI2-THOR are hand-built by professional 3D artists);
> - *Photo-Realistic Targets*: the availability of photo-realistic objects that can be used as navigation targets. In PInNED we carefully selected objects with realistic appearances. Procedurally generated datasets, instead, tend to favor customizability over realism;
> - *Additional Objects*: the inclusion of target objects which were not present at the time of capture. Datasets like GOAT-Bench target objects which were already present in the acquired scene, while PINnED targets objects injected in the scene afterwards;
> - *Visual Reference*: providing visual target references (*i.e.* images) for each navigation episode;
> - *Descriptive Reference*: providing natural language descriptions as targets for each navigation episode;
> - *Variable Placement*: the possibility of having variable spawning positions for the targets within the dataset;
> - *Instance Goal*: the inclusion of navigation episodes in which the goal is to reach the exact instance indicated to the agent.
>
> We will add this description of Table 1 in the revised paper.
>
> **The authors could add the comparison with ProcTHOR, especially regarding object interaction capabilities and category coverage.**
>
> ProcTHOR is a framework built on AI2-THOR to procedurally generate interactive environments, enabling the evaluation of data augmentation and large-scale training in different Embodied AI tasks. PINnED is a dataset designed specifically to study our introduced PIN task, in which the agent is tasked with finding a specific instance according to target images or textual descriptions.
>
> ProcTHOR includes 1,633 instances across 108 object categories, with the ability to vary brightness, colors, materials, and object states. These categories include several household objects, covering generic objects, such as *pen* or *apple*, objects that can be personal, such as *mug* and *watch*, and large objects that are unlikely to change their placement in the environment, such as *fridge*, and *window*. PINnED presents 18 object categories that can be personal, with the specific purpose of accompanying the task in which the agent have to distinguish instances belonging to the same category. All the categories represent objects that can be moved frequently in the environment and do not have a predefined location.
>
> As well as most procedural datasets, ProcTHOR sacrifices realism in favor of interactivity, scalability, and customizability. PINnED, as a task-specific dataset, favors photo-realistic environments and objects. Indeed, it is the first instance-based navigation dataset based on both photo-realistic environments and injected objects, that can be moved frequently and with multimodal targets. Interactivity with the objects is out of scope for this work, however the addition of external objects pave the way for possible future enhancements where interactivity is needed.
>
> We will add this discussion and ProcTHOR in the comparison shown in Table 1. The updated table is available in the attached PDF.
>
> **The authors could provide more details about the distribution logic for the added objects within the environments.**
>
> We thank the Reviewer for raising this concern. The scope of our work is to provide a benchmark to evaluate an agent tasked with finding a specific object that can be located anywhere in an unexplored environment, where distractors of the same category are present; hence, the object categories are selected according to the following criteria: (i) objects that are highly customizable in terms of shapes, colors, and other visual aspects, (ii) objects that are frequently moved and can be placed anywhere, and (iii) objects of common use for which is reasonable to ask a robot to find. Accordingly, we selected a set of plausible surface categories in which these objects could be placed. We recognize that, across the selected 18 categories, there are few object-surface combinations which are less likely than others, however all the combinations are possible in a real-world scenario. Considering a placement procedure constrained on specific categories could represent an interesting improvement for future works, nevertheless the current procedure presents advantages: (i) does not provide any prior knowledge about the possible placement of the object, (ii) provides an evaluation benchmark that includes infrequent situations, and (iii) when the dataset is used for training, improves the robustness against not usual scenarios. We will add this discussion in Sec. E.
>
> **The authors should discuss the scalability of the dataset with the random placement of objects, since it could lead to unrealistic scenarios and potentially infinite scalability.**
>
> We thank the Reviewer for raising this point. To avoid the infinite scalability issue and the presence of an unrealistic number of objects in the environments, we introduced a maximum number of additional objects per episode during the generation of the dataset (more details in Sec. D).
> We underline that the ability to place objects in a wide range of locations represents a strength, since increases the difficulty of the task and challenges the agent to identify the correct instance regardless of appearance and topology of the scene, thereby enhancing generalization capabilities.

---

> > ### Comment · Reviewer_UcJy · 2024-08-17
> > **Thank the authors for the rebuttal**
> >
> > Thank the authors for their rebuttal. However, I still have some concerns, primarily centered around the reasonableness of the way to add personalized objects, which appears to be the core contribution of the work. Specifically, I am concerned about the rationale behind the inclusion and placement of these personalized objects in the navigation environment.
> >
> > First, although the authors mention that PINnED focuses more on photo-realistic environments and objects compared to ProcThor, I am puzzled by the scaling of objects within your dataset. As shown in Figure 2, the far-left image depicts eyeglasses placed on a table that seems disproportionate in size. This raises questions about whether you uniformly scaled all objects according to a fixed ratio or if there was random scaling involved. If scaling was random, how did you determine the appropriate size for objects like teddy bears, which can vary widely in size? Furthermore, it is questionable whether PINnED can still be considered photo-realistic after the introduction of these personalized instances. The images in the supplementary materials suggest that these objects were added in a rather rough manner, which might detract from the overall realism of the environment.
> >
> > Moreover, you claim that "we selected a set of plausible surface categories." I believe this statement requires further clarification and cannot be left vague. For example, it is generally not considered plausible for a teddy bear to be placed on a kitchen table. Could you provide more detailed reasoning behind the selection of object placements and how you ensured they align with realistic scenarios?

---

> > > ### Author Response · Authors · 2024-08-18
> > > **Further details on object scaling, placement and photo-realism**
> > >
> > > We would like to thank again the Reviewer for the time dedicated to our work.
> > >
> > > **Could the authors provide further details about the scaling of the objects within the dataset?**
> > >
> > > We thank the Reviewer for raising this clarifying question. In Section 3.3 of the main paper, we mentioned that objects are manually scaled, without providing further details. In this procedure, we rendered each given object in a scene from HM3D and varied the scale of the object until the result was realistic according to our judgment. Hence, each of the 338 additional objects has a manually fixed scale that is adopted when the object is injected into the navigation episodes. The distribution of the volumes of the object bounding boxes per category is depicted in Figure H of the Appendix. We will add these details on the object scaling procedure to Section 3.3 of the revised version of the main paper.
> > >
> > > **It is questionable whether PInNED can still be considered photo-realistic after the introduction of these personalized instances. Could the authors explain how PInNED can still be considered photo-realistic after the introduction of personalized instances?**
> > >
> > > We thank the Reviewer for highlighting this aspect. The 3D objects have been carefully selected from Objaverse-XL with human supervision, with the criterion of prioritizing photo-realism. This process resulted in most of the objects with meshes acquired from real-world scans because they show an improved visual alignment with the scenes of HM3D. We acknowledge that rendering these objects in HM3D environments may lead to situations in which the object seems *in evidence* with respect to the background because the meshes are acquired from different real-world sources in terms of camera specifications and lighting conditions. However, we point out that our approach produces better results with respect to direct competitors like MultiON and THDA, as it can be observed in Figure 2 of the main paper and Figure A of the Appendix. We will add this explanation to the revised version of the main paper.
> > >
> > > **Could the authors provide more detailed reasoning behind the selection of object placements and how is the alignment with realistic scenarios ensured?**
> > >
> > > We thank the Reviewer for raising the question. The set of selected surfaces is composed of: "*amrchair*", "*bed*", "*bench*", "*cabinet*", "*piano*", "*rug*", "*sofa*", and "*table*". These surfaces are valid for all the object categories and there are no subsets of surfaces dedicated to specific categories. There are categories, especially '*shoes*', that are unlikely to be placed on certain surfaces. However, the scope of the task is to have objects that *could* be placed everywhere and teach a robotic agent to find them. A teddy bear is not necessarily located on the bed, but could be located anywhere, even on the kitchen table. If we assume a real-world scenario in which a child forgets the teddy bear on the kitchen table, the agent should not go directly to the bedroom, but look for the object in the whole environment. This is the reason for which we adopted a consistent spawning mechanism across all the categories. We identify this combination of objects that could be placed everywhere and the consistent spawning mechanisms as the correct approach for providing a dataset covering a large set of possible real-world scenarios that avoid the exploitation of prior knowledge on the object placement. Further details on surfaces are reported in Section E of the Appendix. We will improve this section with the provided answer.

---

> > ### Comment · Reviewer_UcJy · 2024-08-19
> > **Thank the authors for further explanation**
> >
> > Thanks for the author's explanation. I would temporarily raise my recommendation.
> >
> > But what's going on with the glasses in the far-left image of Figure 2? As I mentioned in my first reply, I am curious about it.
> >
> > Moreover, concerning the objects integrated into the environment via photos, I am looking for clarification on the scale factor in the proposed dataset. Does it relate to the dimensions of the agent captured photos, or does it relate to the 3D environment?
> >
> > Additionally, I would appreciate an explanation for the choice of using photographic representations rather than 3D models in the simulation. The lack of orientation adjustment in the added instances when the agent rotates could potentially detract from the realism of the simulation.

---

> > > ### Author Response · Authors · 2024-08-20
> > > **Clarifications on scale and orientation**
> > >
> > > We thank the Reviewer again for all the insightful comments and the clarifying questions.
> > >
> > > **But what's going on with the glasses in the far-left image of Figure 2?**
> > >
> > > We attributed a bounding box to the glasses in the far-left image of Figure 2 of dimensions 16.8 $\times$ 6.8 $\times$ 17.7 cm. We measured two pairs of real eyeglasses. One pair is 13.5 $\times$ 4.0 $\times$ 14.0 cm, while the other is 14.5 $\times$ 6.2 $\times$ 15.0 cm. Probably, the unusual heart-shaped frame of the glasses led us to slightly overestimate the dimensions during the rescaling procedure. Furthermore, in Figure 2 these eyeglasses are placed on the table of a child's room, which is composed of smaller objects. These factors, along with the perspective of the agent, may induce a distorted perception of the scene.
> > >
> > > **Concerning the objects integrated into the environment via photos, I am looking for clarification on the scale factor in the proposed dataset. Does it relate to the dimensions of the agent captured photos, or does it relate to the 3D environment?**
> > >
> > > We thank the Reviewer for the clarifying question. We apologize for the misunderstanding that probably derived from our previous sentence "*rendering these objects in HM3D environments may lead to situations in which the object seems in evidence with respect to the background because the meshes are acquired from different real-world sources in terms of camera specifications and lighting conditions*". The objects that we inject into the environments of HM3D are actually 3D models. Hence, each 3D model has a fixed scale that is manually chosen with the procedure reported in the previous answer, by comparing its size with the surrounding scene in a predetermined HM3D environment. With *rendering* we referred to the step in which the 3D model of the object is actually instantiated in the environment during the simulation. While, when discussing the varying camera specifications and lighting conditions, we are referring to the potential differences between the scans used to create the HM3D environments and those used by the owners of objects contained in the Objaverse-XL dataset to generate the corresponding 3D models.
> > >
> > > **I would appreciate an explanation for the choice of using photographic representations rather than 3D models in the simulation. The lack of orientation adjustment in the added instances when the agent rotates could potentially detract from the realism of the simulation.**
> > >
> > > We apologize again for the misunderstanding. Since the objects are 3D models rather than photographic representations, the simulation is not impacted by the issue regarding the lack of orientation adjustment.

---

> > > > ### Comment · Reviewer_UcJy · 2024-08-29
> > > > **Thank the authors for further explanation**
> > > >
> > > > First, I would like to express my appreciation for the authors' prompt response.
> > > >
> > > > However, I am currently having difficulty understanding the process of injecting personalized objects into the environment. As mentioned in Lines 180 to 186 of the paper: "The input images for each target personalized object are generated by rendering the 3D mesh of the object in an isolated setting. Specifically, the input images are not expected to match the camera specification of the navigating agent." It appears that the authors are using images instead of directly injecting 3D models into the environment. Could you please clarify this point and provide the detailed code for 3D model injection? I have only found examples of image input in the provided GitHub repository, and I am keen to understand the specific method for implementing 3D model injection.

---

> > > > > ### Author Response · Authors · 2024-08-30
> > > > > **Clarification on 3D model rendering**
> > > > >
> > > > > **I am currently having difficulty understanding the process of injecting personalized objects into the environment. As mentioned in Lines 180 to 186 of the paper: "The input images for each target personalized object are generated by rendering the 3D mesh of the object in an isolated setting. Specifically, the input images are not expected to match the camera specification of the navigating agent." It appears that the authors are using images instead of directly injecting 3D models into the environment. Could the authors clarify this point and provide the detailed code for 3D model injection? I have only found examples of image input in the provided GitHub repository, and I am keen to understand the specific method for implementing 3D model injection.**
> > > > >
> > > > > We thank the Reviewer for raising the question. We would like to clarify that this question comprises two different phases: (i) the phase of creation of the input reference images, that are used to specify the target object to the agent, and (ii) the phase of injection of the 3D models of the additional objects in the environments of Habitat. Lines 180 to 186 of the main paper refer to the creation of the reference images. In this phase, the 3D objects are rendered in an isolated setting and three images are acquired while rotating the object. These images will be used as visual references representing the goal and provided to the agent at the beginning of each navigation episode. The example images in the GitHub repository are these reference images. On the other side, the phase of injecting 3D objects is executed during the initialization of the Habitat environment used to perform each specific navigation episode. In this phase, the actual 3D models of each additional object are resized according to manual annotations contained in the repository inside ```data/datasets/pin/hm3d/v1/objects/``` and are placed on top of the suitable surfaces as mentioned in the main paper and the previous answers. The code that performs this operation can be found at ```pin/habitat/core/env.py``` at Line 271 in the GitHub repository. Following, we report the code snippet:
> > > > > ```
> > > > >     object_id = str(self.current_episode.goals[0].object_id)
> > > > >     current_goal = self.current_episode.goals[0].object_category
> > > > >     dataset_index = self.object_to_dataset_mapping[current_goal]
> > > > >     obj_handle_list = obj_templates_mgr.get_template_handles(object_id)[0]
> > > > >     object_box = rigid_obj_mgr.add_object_by_template_handle(
> > > > >         obj_handle_list, light_setup_key=NO_LIGHT_KEY,
> > > > >     )
> > > > >
> > > > >     pos = np.array(self.current_episode.goals[0].position)
> > > > >     object_box.translation = np.array(pos)
> > > > > ```
> > > > > Specifically, the ```obj_templates_mgr``` loads the manual annotation of the provided ```object_id```, which contains the path to the 3D model, the resizing scale, and the semantic category. Then, this annotation is provided to the ```rigid_obj_mgr``` which is in charge of instantiating the 3D model of the object accordingly. Finally, we use this pointer to move the object to its assigned position as defined in the episode annotation.

---

> > ### Comment · Reviewer_UcJy · 2024-09-01
> > **Thank the authors for further explanation**
> >
> > Thank the authors for their reply. I now understand your points.
> >
> > However, I still have doubts about the influence of the reference image on the results. If the reference image has a big impact, it's hard to call this a personalized instance based navigation because the image itself is specific to the instance. The proposed task seems more like image-based navigation. I hope the authors can show that their baselines can be personalized without just using specific images.
> >
> > I am also curious why the authors think that using images of specific instances can still be seen as personalized, since they do not use the concept of categories when the specific images are provided.
> >
> > Considering that the discussion time is about to end, I will not adjust my score due to the lack of a response from the authors.

---

> > > ### Author Response · Authors · 2024-09-01
> > > **Clarification on personalization**
> > >
> > > **However, I still have doubts about the influence of the reference image on the results. If the reference image has a big impact, it's hard to call this a personalized instance based navigation because the image itself is specific to the instance. The proposed task seems more like image-based navigation. I hope the authors can show that their baselines can be personalized without just using specific images.**
> > >
> > > We thank the Reviewer for raising the interesting point. In our task PIN, the target object is indicated to the robotic agent through a set of reference images or textual descriptions. The reference images, as mentioned in the previous answers, are acquired by rendering the 3D model of the object in an isolated context and rotating the model. Hence, when the agent navigates to find the object according to the reference images, our task belongs to the field of image-based navigation. However, it presents several differences from previous tasks, as reported in Sec. 3.2 of the main paper and in Sec. D of the Appendix. The most similar task is InstanceImageNav, in which a reference image of the target object, in the exact context in which it is placed in the environment, is provided to the agent. In our task, the reference images represent the target object in a different context from the one in which it is placed since our introduced objects are objects that can be moved frequently. These characteristics disable the possibility of exploiting the surrounding context. In this scenario, key-point matching baselines, such as SuperGlue (Tab. 2 of the main paper) and IEVE (answer to Reviewer 94wz), perform worse than semantic-based approaches, such as DINO and CLIP. However, in semantic-based approaches, the concept of personalization is crucial, since their feature embeddings must encode both the general category and the fine-grained properties, such as colors and textures, of the object to be able to distinguish it from the same-category distractors. To analyze this behavior, we introduce the set of metrics discussed in Sec. 5.1 of the main paper.
> > >
> > > **I am also curious why the authors think that using images of specific instances can still be seen as personalized, since they do not use the concept of categories when the specific images are provided.**
> > >
> > > We refer to our work as a personalization task since the agent is tasked with finding personal instances of highly customizable categories while there are other distractors of the same category in the environment. Despite the concept of category not directly provided to the agent, the distractors of the same category represent one of the main obstacles in accomplishing the task, especially in semantic-based approaches as mentioned in the previous answer. Hence, the proposed benchmark is focused on the capabilities of the agent in recognizing the correct instances along the same-category distractors.

---

### Official Review · Reviewer_94wz · 2024-08-04

**Rating:** 7
**Confidence:** 3
**Correctness:** Yes.
**Clarity:** The paper is easy-to-follow.

**Review:**

-	The dataset consists of adding 338 objects to the simulator, which is a good contribution.
-	The introduced Personalized Instance-based Navigation (PIN) task is particularly useful in fine-grained scenarios where multiple instances of the same object category are present in a room.
-	The benchmark was done partially satisfactory, utilizing two settings: a two-stage approach (one with object recognition and one with navigation) and end-to-end navigation baselines.

**Strengths:**

Adding objects from Objaverse-XL to Habitat simulator is beneficial for improving robustness for fine-grained object recognition.
However:
While the authors did a great job differentiating between PInNED and GOAT-bench, there are still major concerns regarding the evaluation. Specifically, there is a lack of comparison between the performance of baselines between two benchmarks. For example, it would be insightful to see the performance of baselines trained with two types of descriptions: one with fine-grained details and the other with general descriptions (such as "black and white striped trekking backpack" vs. "backpack" as mentioned in L147). A thorough experiment addressing this limitation would be beneficial, as it would highlight the strength of training with fine-grained descriptions.

**Additional Feedback:**

See limitation and improvements. I would raise my score if my major concerns are addressed.

**Documentation:**

The dataset is available and well-documented.

**Ethics:**

No.

**Limitations:**

The limitations and broader impact of the work are thoroughly discussed.

**Opportunities For Improvement:**

The evaluation results of baselines seem poor. The authors should consider implement the following state-of-the-art ImageGoal baselines:
[1] Lei, X., Wang, M., Zhou, W., Li, L., & Li, H. (2024). Instance-aware Exploration-Verification-Exploitation for Instance ImageGoal Navigation. In CVPR 2024.

**Relation To Prior Work:**

The paper clearly discusses the prior work, GOAT-bench, which has similar contributions. However, some details are still missing.

**Summary And Contributions:**

This paper presents the PInNED dataset, an extension of Habitat simulator using HM3D scenes and objects from Objaverse-XL. This dataset facilitates object navigation tasks by providing references such as images or descriptions of the target object.

---

> ### Author Rebuttal · Authors · 2024-08-16
>
> We thank the Reviewer for the time devoted to our paper, for providing valuable comments, and for acknowledging the utility of the proposed dataset and benchmark. In the following, we address the specific concerns pointed out by the Reviewer.
>
> **It would be insightful to see the performance of baselines trained with two types of descriptions: one with fine-grained details and the other with general descriptions (such as "black and white striped trekking backpack" vs. "backpack").**
>
> We agree that comparing the performance of the baselines using different types of descriptions would significantly enhance the value of the proposed dataset. To address this, we have conducted an ablation study, evaluating the baselines with both fine-grained and general object categories. The results of this analysis are summarized in Table A provided in the attached PDF.
>
> Specifically, we conducted the following experiments. For the modular agents based on CLIP and OWL as the matching module, we leveraged the general object category (e.g. *backpack*) instead of the fine-grained textual descriptions as navigation targets, while maintaining the same similarity threshold. The results on both CLIP and OWL present similar behaviors: the number of successful episodes is slightly increased, but also the number of episodes in which the agent mistakes reaching distractors of the same category and the number of matches with them increased. Moreover, the reduction in the average number of steps indicates that similarities, on average, are higher. The increase in successful episodes is surprising but in line with the findings of previous works in the literature [A, B], which demonstrate that current vision-language models struggle with fine-grained details. These results show that our work can help future works in the realization and evaluation of vision-language models with improved understanding capabilities of details.
>
> Concerning the end-to-end agent RIM, we trained the model on the CLIP embeddings extracted from the general object categories instead of the fine-grained textual descriptions. The results show a lower number of successful episodes and a higher number of episodes in which the agent reaches a distractor of the same category. We will add these experiments in the revised version of the paper.
>
> [A] Bianchi L., Carrara F., Messina N., Gennaro C., & Falchi F., "*The devil is in the fine-grained details: Evaluating open-vocabulary object detectors for fine-grained understanding*" in CVPR 2024.
>
> [B] Bravo M. A.,  Mittal S., Ging S., & Brox T., "*Open-vocabulary Attribute Detection*", in CVPR 2023.
>
> **The authors should consider implementing the following state-of-the-art ImageGoal baselines:
> [C] Lei, X., Wang, M., Zhou, W., Li, L., & Li, H., "*Instance-aware Exploration-Verification-Exploitation for Instance ImageGoal Navigation*" in CVPR 2024.**
>
> We thank the reviewer for the suggestion. At the time of submission, we did not evaluate the referenced method as it was not yet published and the source code had not yet been released. Nonetheless, we have evaluated the method proposed in [C] (*i.e.* IEVE), and the results are reported in Table B in the attached PDF. We will integrate this experiment into the revised version of the main paper.
>
> To evaluate IEVE on PIN, we adapted some components to match the different requirements of our task. Specifically, we first collected an auxiliary dataset composed of, for each goal in the training set, 10 positive samples and one negative sample, containing a distractor of the same category of the goal. We trained InternImage to classify the 18 categories of our dataset using the goal images of the training set. Instead of the InternImage segmentation model, since, to the best of our knowledge, no segmentation dataset contains all our categories, we adopted the open-vocabulary segmenter GroundedSAM. For the image-matching step, we exploited LightGlue on the keypoints extracted with DISK as in the original IEVE paper.
>
> The results show an improvement with respect to the other image-matching modular agent, based on SuperGlue. This is motivated by the fact that IEVE, differently from other image-matching approaches, combines LightGlue with a semantic detector, allowing the agent to focus only on observations that contain objects of the target category. This behavior is confirmed by the increased numbers of target matches, category matches, and category errors. Future works should consider similar approaches in which advantages from both semantic and keypoint-based matching are exploited.
>
> **Some details are still missing in the comparison with prior work.**
>
> We thank the Reviewer for pointing this out. We will integrate a more detailed description of the dataset comparison into the revised version of the paper, along with additional experiments. These updates will also address the points raised by the other reviewers to ensure a more comprehensive comparison with prior work. In particular, we will add to the revised version of the paper: (i) results and discussion on the IEVE baseline, (ii) a more detailed explanation of the content and columns in the comparison of datasets contained in Table 1 of the main paper (answer to Reviewer UcJy), (iii) comparison with the ProcTHOR dataset (answer to Reviewer UcJy), and (iv) discussion on how the similarity of objects influence the performance (answer to Reviewer 6hP2). We are available to provide the Reviewer with further clarifications during the author/reviewer discussion period.

---

### Author Rebuttal · Authors · 2024-08-17

We sincerely thank the Area Chair and Reviewers for the time they devoted to our work, for their insightful comments, and for their valuable suggestions.
Based on the Reviewers’ feedback, we have addressed their concerns and carefully revised our main paper.
Specifically, we have provided detailed responses to each reviewer, elaborating on their concerns and incorporating the suggested improvements to enhance the clarity and quality of our work. In order, we discussed:
- The performance of the baselines when considering the general object category instead of the fine-grained descriptions that we provided in the PINnED dataset for each target object;
- The results of the IEVE approach adapted to the PIN task, its comparison with the other baselines, and further details in the comparison with prior works;
- A more detailed explanation of the content of Table 1, in which we compared PIN with prior datasets and benchmarks in the object-driven navigation literature;
- The comparison with ProcTHOR, focusing on interaction capabilities and category coverage;
- The distribution logic that we adopted for injecting the objects within the environments of the PINnED dataset;
- The scalability of the dataset with the adopted approach for the placement of the injected objects;
- Additional information about the experimental setup and methodology;
- The novelty of our work and potential real-world applications it implies;
- How real-world navigation scenarios with dynamic environments and varying user preferences are included in our dataset and benchmark;
- Diversity and bias of personal objects and environments, ethical concerns regarding privacy and data security, data protection and potential negative societal impacts of our work;
- How the similarity of objects influences the performance and how it can be used to improve the performance.

---

### Decision · Program_Chairs · 2024-09-26

**Decision:**

Accept (Poster)

**Comment:**

The paper proposes the Personalized Instance-based Navigation (PIN) task where agents need to navigate to a specific instance of an object (specified by an image + text description) that may be similar to distractor objects.  To study the task, a dataset (PInNED) of 865.5k training episodes and 1.2k validation episodes are provided.  These episodes are built on top of scenes from HM3D semantics and 338 objects from Objaverse-XL.  Experiments are conducted comparing the performance of common embodided navigation baselines on the proposed dataset.

All four reviewers were positive on the work and advocates for acceptance.  Reviewers found the paper to be well-written, with the task clearly differentiated from prior navigation tasks.

While there has been prior work that place objects in scenes to create episodes for embodied navigation, the AC agrees that the task is well articulated and dataset can be of potential interest to the community.  The AC recommends acceptance and encourages to improve the paper for the camera ready based on feedback from the reviewers, including providing clarifying details on comparison with prior work, details about the placement logic for objects, experimental setup and methodology, discussion of the limitations of the work, etc.